# Solving the 2-norm k-hyperplane clustering problem via multi-norm formulations

**Stefano Coniglio**
Department of Economics
University of Bergamo
Bergamo, Italy
`stefano.coniglio@unibg.it`

## Abstract

We propose a method to solve $k$-HC$_2$—the $k$-Hyperplane Clustering problem that asks to find $k$ hyperplanes that minimize the sum of squared 2-norm (Euclidean) distances between each point and its closest hyperplane—to global optimality via spatial branch-and-bound (SBB) techniques. Our method strengthens a mixed integer quadratically-constrained quadratic programming formulation for $k$-HC$_2$ with constraints that arise when formulating the problem in $p$-norms with $p \neq 2$. In particular, we show that, for every (suitably scaled) $p \in \mathbb{N} \cup \{\infty\}$, one obtains a variant of $k$-HC$_2$ whose optimal solutions yield lower bounds within a multiplicative approximation factor. We focus on the case of polyhedral norms where $p = 1, \infty$ (which are disjunctive-programming representable), and prove that strengthening the original formulation by including, on top of its 2-norm constraints, the constraints of one of the polyhedral norms leads to an SBB method where nonzero lower bounds are obtained in a a number of nodes that is linear in $n$ and $k$ (rather than exponential). Experimentally, our method leads to very large speedups, reducing median solve times by up to 41× while increasing the total number of solved instances by up to 63%, drastically improving the problem's solvability to global optimality.

## 1 Introduction

Given $m$ points $\{a_1, \ldots, a_m\}$ in $\mathbb{R}^n$, the *$k$-Hyperplane Clustering* problem, or $k$-HC$_2$, asks for identifying $k$ hyperplanes which minimize the sum of the squares of the distances between each point and the hyperplane closest to it in Euclidean (2-norm) distance. $k$-HC$_2$ arises when relationships of *co-linearity* (in $\mathbb{R}^2$) or *co-(hyper)planarity* (in $\mathbb{R}^n$) are sought. One of the problem's most natural applications is line/surface detection in digitally-sampled images and in 3d environments Amaldi & Mattavelli (2002). More applications are found in diverse areas such medical prognosis Bradely & Mangasarian (2000), linear facility location Megiddo & Tamir (1982), discrete-time piecewise affine hybrid system identification Ferrari-Trecate et al. (2003), piecewise-affine model fitting Amaldi et al. (2016), principal/sparse component analysis Washizawa & Cichocki (2006); He & Cichocki (2007); Tsakiris & Vidal (2017), nonlinear regression He & Qin (2010), dictionary learning Zhang et al. (2013), LiDAR data classification Kong et al. (2013), and sparse matrix representation Georgiev et al. (2007).

$k$-HC$_2$ was first introduced by Bradely & Mangasarian (2000), where it is shown that, with $k = 1$, the problem is solved by computing an eigenvalue-eigenvector pair of a suitably defined matrix built as a function of the data points. $k$-HC$_2$ is $\mathcal{NP}$-hard in any norm since fitting $m$ points in $\mathbb{R}^n$ with $k$ hyperplanes with 0 error is $\mathcal{NP}$-complete even for $n = 2$ (Megiddo & Tamir, 1982). To tackle $k$-HC$_2$ when $k \geq 2$ without optimality guarantees, Bradely & Mangasarian (2000) proposed an adaptation of the popular $k$-means heuristic by MacQueen (1967). An exact Mixed Integer Quadratically Constrained Quadratic Programming (MI-QCQP) formulation is proposed by Amaldi & Coniglio (2013), together with a heuristic for larger-scale instances. Works addressing variants of $k$-HC$_2$ asking for the smallest number of hyperplanes with a distance no larger than a given $\epsilon > 0$ are found in Dhyani & Liberti (2008); Amaldi et al. (2013).

**Contributions.** We propose a method to solve $k$-$\mathrm{HC}_2$ to global optimality via a spatial branch-and-bound (SBB) technique. We strengthen a classical mixed-integer quadratically-constrained quadratic programming (MI-QCQP) formulation for $k$-$\mathrm{HC}_2$ by including constraints (and variables) that arise when formulating the problem in another $p$-norm ($p \neq 2$). We show that, under mild assumptions, the inclusion of constraints stemming from a version of $k$-$\mathrm{HC}_2$ formulated in one of the two polyhedral norms (where $p = 1, \infty$) leads to an SBB method where a nonzero global lower bound is obtained in a linear number of SBB nodes, as opposed to the exponential number that is necessary when the classical formulation is used. Our experiments reveal that our method leads to very large speedups, reducing median solve times by up to 41× while increasing the total number of solved instances by up to 63%, substantially improving the problem's solvability to global optimality.

## 2 PRELIMINARIES

Given a point $a \in \mathbb{R}^n$, its $p$-norm with $p \in \mathbb{N} \cup \{\infty\}$ is $\|a\|_p := \lim_{q \to p} \left( \sum_{h=1}^n |a_h|^q \right)^{1/q}$. In particular, for $p = 1, 2,$ and $\infty$ we have $\|a\|_1 = \sum_{h=1}^n |a_h|$, $\|a\|_2 := \left( \sum_{h=1}^n |a_h|^2 \right)^{1/2}$, and $\|a\|_\infty = \max_{h \in [n]} \{|a_h|\}$.[1] The $p$-norm point-to-hyperplane distance $d_p(a, H)$ between a point $a \in \mathbb{R}^n$ and a hyperplane $H := \{x \in \mathbb{R}^n : x^\top w = \gamma\}$ of parameters $(w, \gamma) \in \mathbb{R}^{n+1}$ is defined as the $p$-norm distance between $a$ and the point $y \in H$ that is closest to it. Namely, $d_p(a, H) := \min_{y \in H} \|a - y\|_p$. Different arguments, including Lagrangian duality—see Mangasarian (1999), can be used to show that $d_p(a, H) = \frac{|w^\top a - \gamma|}{\|w\|_{p'}}$, where $p$ and $p'$ satisfy $\frac{1}{p} + \frac{1}{p'} = 1$.[2] For $p = 2$, $d_p(a, H)$ is called *Euclidean point-to-hyperplane* (or *orthogonal*) *distance*. In many applications, such a distance is preferred as it leads to solutions that are invariant to rotations of the data points.

In spite of being defined on top of a $p$-norm, the distance function $d_p$ is intrinsically nonconvex w.r.t. $w$ regardless of the choice of $p$ (the proof is in the appendix):

**Proposition 1.** *Given a hyperplane $H := \{x \in \mathbb{R}^n : x^\top w = \gamma\}$ and a point $a \in \mathbb{R}^n$, the function $d_p(a, H) = \frac{|w^\top a - \gamma|}{\|w\|_{p'}}$, where $\frac{1}{p} + \frac{1}{p'} = 1$, is a nonconvex function of $(w, \gamma)$ for every $p \in \mathbb{N} \cup \{\infty\}$.*

This makes $k$-$\mathrm{HC}_2$ substantially harder than classical machine learning problems where a norm is minimized, and motivates the adoption of SBB techniques for solving it to global optimality.

## 3 APPROXIMATING $k$-$\mathrm{HC}_2$ USING DIFFERENT NORMS

Given $m$ points $\{a_1, \ldots, a_m\}$ in $\mathbb{R}^n$, the most compact nonlinear programming (NLP) formulation for $k$-$\mathrm{HC}_2$ reads:[3]

$$(k\text{-}\mathrm{HC}_2) \qquad \min_{(w,\gamma)} \left\{ \sum_{i=1}^m \min_{j \in [k]} \left\{ \frac{(a_i^\top w_j - \gamma_j)^2}{\|w_j\|_2^2} \right\} \right\},$$

where $(w_j, \gamma_j) \in \mathbb{R}^{n+1}$, $j \in [k]$, are the hyperplane parameters. ($k$-$\mathrm{HC}_2$) has a non-smooth objective function due to Proposition 1 and, since $\|w_j\|_2^2 = w_j^\top w_j$, it features ratios of quadratics. While the inner $\min$ operator can be easily dropped by introducing binary assignment variables (see below), this formulation is unsuitable for most nonlinear programming solvers as the denominator vanishes when $w_j = 0$.

In the remainder of the paper, we will study $k$-$\mathrm{HC}_{(p,c)}$, a generalized version of $k$-$\mathrm{HC}_2$ which employs a $p$ norm not necessarily equal to 2 and which is parametric in a constant $c \geq 0$. Its NLP formulation, where $\frac{1}{p} + \frac{1}{p'} = 1$, reads:

$$(k\text{-}\mathrm{HC}_{(p,c)}) \qquad \min_{(w,\gamma)} \left\{ \sum_{i=1}^m \min_{j \in [k]} \left\{ (a_i^\top w_j - \gamma_j)^2 \right\} : \|w_j\|_{p'}^2 \geq c, j \in [k] \right\},$$

---

[1] Throughout the paper, we adopt the notation $[\xi] := 1, \ldots, \xi$ for every $\xi \in \mathbb{N}$.

[2] Two norms where $\frac{1}{p} + \frac{1}{p'} = 1$ are called *dual*. The 2-norm is self dual and the 1 and $\infty$-norms are dual.

[3] We report mathematical programming formulations in brackets and optimization problems without them.

Letting, for a problem $P$, $\mathrm{OPT}(P)$ be its optimal solution value, the validity of ($k$-$\mathrm{HC}_{(p,c)}$) and the role that $c$ plays in it are shown by the following lemma (the proof is in the appendix):

**Lemma 1.** *The solutions to ($k$-$\mathrm{HC}_{(2,1)}$) and ($k$-$\mathrm{HC}_2$) coincide. Also, ($k$-$\mathrm{HC}_{(p,c)}$) is quadratically homogeneous w.r.t. $c$, i.e., $\mathrm{OPT}(k\text{-}\mathrm{HC}_{(p,c)}) = c^2\,\mathrm{OPT}(k\text{-}\mathrm{HC}_{(p,1)})$.*

The property shown by the lemma will be useful to guide our choice of which $p$ we should use to introduce additional norm constraints to the formulation of $k$-$\mathrm{HC}_2$ (which, we recall, is the version of the problem that we aim to solve in this paper) in order to strengthen it.

**Rationale.** Investigating $k$-$\mathrm{HC}_{(p,c)}$ with $(p,c) \neq (2,1)$ is of interest for two reasons. First, as shown in this section, doing so allows us to show that, for a suitable choice of $p$ and $c$, the optimal solutions to $k$-$\mathrm{HC}_{(p,c)}$ are approximate solutions (to within an approximation factor) of those to $k$-$\mathrm{HC}_{(2,1)}$. Second, as shown in the next two sections, the study of $k$-$\mathrm{HC}_{(p,c)}$ allows us to prove that, again for a suitable choice of $p$ and $c$, the formulations ($k$-$\mathrm{HC}_{(p,c)}$) and ($k$-$\mathrm{HC}_{(2,1)}$) can be intersected to obtain a *strengthened formulation* which is valid for $k$-$\mathrm{HC}_2$ and which is also much easier to solve both in theory and practice.

**Novelty.** While changes of norm are frequent in the ML literature, the dual norm in the denominator of the point-to-hyperplane distance requires, for our results, switching between primal and dual norms and applying suitable scaling factors to the problem's constraints in a way that, to our knowledge, is new. The idea of *intersecting* formulations derived for different norms, which leads to a provably tighter approximation factor, is also, to our knowledge, uncommon in the literature. We also manage to establish lower bounds on the number of branching operations needed to compute a nonzero lower bound (after which pruning becomes possible), a type of result which is extremely rare in integer programming (let alone nonlinear integer programming).

## 3.1 The general case

We show that, whichever version of $k$-$\mathrm{HC}_{(p,c)}$ one aims to solve (be it the 2-norm one with $c = 1$ or another one), the optimal-solution value of $k$-$\mathrm{HC}_{(q,c')}$ for *any* choice of $q$ and a suitable $c'$ is within an approximation factor of the optimal-solution value of $k$-$\mathrm{HC}_{(p,c)}$:

**Theorem 1.** *Let $p,q \in \mathbb{N} \cup \{\infty\}$ and $c > 0$. The three positive scalars $\alpha(p,q), \beta(p,q), \delta(p,q)$ which, for all $x \in \mathbb{R}^n$, satisfy the congruence inequality $\alpha(p,q)\|x\|_p \leq \beta(p,q)\|x\|_q \leq \delta(p,q)\|x\|_p$ for $p,q \in \mathbb{N}\cup\{\infty\}$ also satisfy the optimal-value inequality $\frac{\alpha(p,q)^2}{\delta(p,q)^2}\,\mathrm{OPT}(k\text{-}\mathrm{HC}_{(p,c)}) \leq \mathrm{OPT}\left(k\text{-}\mathrm{HC}_{(q,c\frac{\beta(p,q)}{\delta(p,q)})}\right) \leq \mathrm{OPT}(k\text{-}\mathrm{HC}_{(p,c)})$.*

Theorem 1 shows that the optimal solution value of $k$-$\mathrm{HC}_{(q,c')}$ with $c' = c\frac{\beta(p,q)}{\delta(p,q)}$ is a lower bound on the optimal solution value of $k$-$\mathrm{HC}_{(p,c)}$ to within an approximation factor of $\frac{\alpha(p,q)^2}{\delta(p,q)^2}$. This is crucial, as it shows which value to pick for $c'$ for *any* $q$-norm we may choose to obtain a relaxation of $k$-$\mathrm{HC}_{(p,c)}$ and, in particular, one of $k$-$\mathrm{HC}_{(2,1)}$ (which is, ultimately, the problem we aim to solve).

We remark that Theorem 1 can be extended to produce an approximation of $k$-$\mathrm{HC}_{(p,c)}$ from above to within an approximation factor—we omit the details since, here, we solely are interested in approximations from below to build tighter relaxations suitable for an SBB method.

Theorem 1 has a nice geometrical interpretation in terms of the feasible regions of ($k$-$\mathrm{HC}_{(p,c)}$) and ($k$-$\mathrm{HC}_{(q,c\frac{\beta(p,q)}{\delta(p,q)})}$). Indeed, with $c' = c\frac{\beta(p,q)}{\delta(p,q)}$, the feasible region of the $q$-norm constraints featured in $k$-$\mathrm{HC}_{(q,c')}$ is a relaxation of the region that is feasible for the $p$-norm constraints of $k$-$\mathrm{HC}_{(p,c)}$. An illustration is reported in Figure 1 for $p = 2, c = 1$ and adopting $q = 1$ (left) and $q = \infty$ (right), for which we have, respectively, $c' = 1$ and $c' = \frac{1}{\sqrt{n}}$.

## 3.2 The case of polyhedral norms with $q = 1, \infty$

We now focus on *polyhedral* norms ($q = 1, \infty$). These are of computational interest due to their tractability: while the constraints $\|w_j\|_q \geq c'$, $j \in [k]$, with $q = 1, \infty$, are nonconvex, they can be stated as disjunctions over polyhedra, thus being mixed-integer-linear-programming representable.

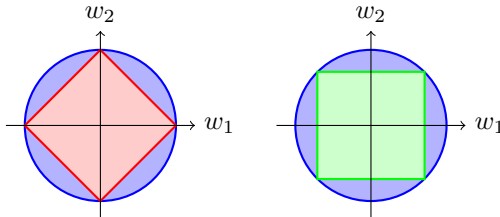

Figure 1: Complements of the feasible regions of $\{w \in \mathbb{R}^2 : ||w||_1 \geq 1\}$ and $\{w \in \mathbb{R}^2 : ||w||_\infty \geq \frac{1}{\sqrt{2}}\}$.

In light of this, we consider the following two relaxations of $k$-HC$_{(2,1)}$ (see again Figure 1 for an illustration of the projection of the feasible regions of these two problems onto the $w$ space for $k = 1$):

$$(k\text{-HC}_{(\infty,1)}) \qquad \min_{(w,\gamma)} \left\{ \sum_{i=1}^{m} \min_{j \in [k]} \left\{ (a_i^\top w_j - \gamma_j)^2 \right\} : \ \|w_j\|_1 \geq 1, j \in [k] \right\},$$

$$(k\text{-HC}_{(1,\frac{1}{\sqrt{n}})}) \qquad \min_{(w,\gamma)} \left\{ \sum_{i=1}^{m} \min_{j \in [k]} \left\{ (a_i^\top w_j - \gamma_j)^2 \right\} : \ \|w_j\|_\infty \geq \frac{1}{\sqrt{n}}, j \in [k] \right\}.$$

Notice that, due to norm duality, ($k$-HC$_{(\infty,1)}$) features 1-norm constraints while ($k$-HC$_{(1,\frac{1}{\sqrt{n}})}$) features $\infty$-norm ones. For these two problems, Theorem 1 leads to the following result (the proof is in the appendix):

**Corollary 1.** $k$-HC$_{(\infty,1)}$ and $k$-HC$_{(1,\frac{1}{\sqrt{n}})}$ satisfy:

$$\frac{1}{n} \mathrm{OPT}(k\text{-HC}_{(2,1)}) \leq \mathrm{OPT}(k\text{-HC}_{(\infty,1)}) \leq \mathrm{OPT}(k\text{-HC}_{(2,1)})$$

$$\frac{1}{n} \mathrm{OPT}(k\text{-HC}_{(2,1)}) \leq \mathrm{OPT}(k\text{-HC}_{(1,\frac{1}{\sqrt{n}})}) \leq \mathrm{OPT}(k\text{-HC}_{(2,1)}).$$

With the first chain of inequalities, the corollary shows that solving $k$-HC$_{(\infty,1)}$, i.e., formulating $k$-HC with the constraint $||w_j||_1 \geq 1$ for all $j \in [k]$, leads to a relaxation to within a $\frac{1}{n}$ approximation factor. With the second one, the corollary shows that solving $k$-HC$_{(1,\frac{1}{\sqrt{n}})}$, i.e., solving the version of $k$-HC with the constraint $||w_j||_\infty \geq \frac{1}{\sqrt{n}}$ for all $j \in [k]$, leads to another relaxation also to within the same approximation factor $\frac{1}{n}$.

### 3.3 Multi-norm relaxation

Since both $||w_j||_1 \geq 1, j \in [k]$, and $||w_j||_\infty \geq \frac{1}{\sqrt{n}}, j \in [k]$, are relaxations of $||w_j||_2 \geq 1, j \in [k]$, a strengthened relaxation of $k$-HC$_{(2,1)}$ can be obtained by simultaneously imposing both. Such a *multi-norm* relaxation, which we refer to as $k$-HC$_{(\text{multi},1)}$, reads

$$(k\text{-HC}_{(\text{multi},1)}) \qquad \min_{(w,\gamma)} \left\{ \sum_{i=1}^{m} \min_{j \in [k]} \left\{ (a_i^\top w_j - \gamma_j)^2 \right\} : \ \begin{array}{ll} \|w_j\|_1 \geq 1, & j \in [k] \\ \|w_j\|_\infty \geq \frac{1}{\sqrt{n}}, & j \in [k] \end{array} \right\}.$$

Letting $||w||_{\text{multi}} := \min\{||w||_1, \sqrt{n}||w||_\infty\}$, one can see that simultaneously imposing $||w_j||_1 \geq 1$ and $||w_j||_\infty \geq \frac{1}{\sqrt{n}}, j \in [k]$, coincides with imposing $||w_j||_{\text{multi}} \geq 1, j \in [k]$. A depiction of the corresponding feasible region is reported in Figure 2.

So far, our analysis has hinged on the possibility of translating a $p'$-norm constraint into the corresponding $d_p$ distance, on which we applied Theorem 1. Deriving an approximation factor for $k$-HC$_{(\text{multi},1)}$ is not straightforward, though. This is because, while vector norms are convex and convex functions have convex sublevel sets, the sub-level sets of the function $||w||_{\text{multi}}$ are not convex and, thus, there is no $p$-norm, $p \in \mathbb{N} \cup \{\infty\}$, whose adoption directly leads to $k$-HC$_{(\text{multi},1)}$.

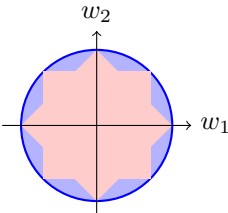

Figure 2: Complement of the feasible region of $\{w \in \mathbb{R}^2 : ||w||_{\text{multi}} \geq 1\}$.

In spite of this, in the following we show that we can still derive an approximation factor by constructing the norm that is implicitly minimized when $\min\{||w||_1, \sqrt{n}||w||_\infty\} \geq 1$ is imposed.

We start with the following lemma (the proof is in the appendix), which shows what combination of point-to-hyperplane distances is minimized in $k$-HC when imposing $\min\{||w||_1, \sqrt{n}||w||_\infty\} \geq 1$:

**Lemma 2.** *Solving $k$-HC subject to $\min\{||w||_1, \sqrt{n}||w||_\infty\} \geq 1$ coincides with solving an unconstrained version of $k$-HC where the point-to-hyperplane distance between $a_i$ and $H_j$ is defined as $\max\{d_\infty(a_i, H_j), \frac{1}{\sqrt{n}}d_1(a_i, H_j)\}$.*

We now prove two new lemmas (the proofs are in the appendix) that show that the function $\max\{||x||_\infty, \frac{1}{\sqrt{n}}||x||_1\}$ is a norm and construct a congruence inequality for it:

**Lemma 3.** *The function $\max\{d_\infty(a_i, H_j), \frac{1}{\sqrt{n}}d_1(a_i, H_j)\}$ is a distance induced by the norm $\max\{||x||_\infty, \frac{1}{\sqrt{n}}||x||_1\}$.*

**Lemma 4.** *The norm $\max\{||x||_\infty, \frac{1}{\sqrt{n}}||x||_1\}$ satisfies the congruence inequality*

$$n^{-\frac{1}{4}}\,||x||_2 \leq \max\left\{||x||_\infty, \frac{1}{\sqrt{n}}||x||_1\right\} \leq ||x||_2.$$

Crucially, the following holds:

**Corollary 2.** *Combining Lemma 4 with Theorem 1, the multi-norm relaxation $k$-HC$_{(\text{multi},1)}$ satisfies*

$$\frac{1}{\sqrt{n}}\,\text{OPT}\big(k\text{-HC}_{(2,1)}\big) \leq \text{OPT}\big(k\text{-HC}_{(\text{multi},1)}\big) \leq \text{OPT}\big(k\text{-HC}_{(2,1)}\big).$$

## 4 SOLVING THE STRENGTHENED FORMULATIONS OF $k$-HC$_{(2,1)}$ VIA SBB

We now focus on solving $k$-HC$_{(2,1)}$ to global optimality via SBB. We analyze the number of SBB nodes needed to compute a nonzero global lower bound when solving a basic formulation of the problem, and then prove that intersecting the basic formulation for $k$-HC$_{(2,1)}$ with one of our relaxations involving polyhedral norms allows for computing a nonzero global lower bound much earlier.

### 4.1 SPATIAL BRANCH-AND-BOUND

The basic idea of the spatial branch-and-bound (SBB) method is to build a dual bound by optimizing over a convex (typically polyhedral) envelope $\text{conv}(F)$ of the feasible region $F$ of the problem. $F$ is then split into two sub-regions $F_1$ and $F_2$ with tighter bounds on at least one variable. This allows for constructing tighter convex envelopes of $F_1$ and $F_2$ in such a way that the optimal solution over $\text{conv}(F)$ is cut off because it does not belong to $\text{conv}(F_1) \cup \text{conv}(F_2)$. $F_1$ and $F_2$ are then recursively optimized in a classical *divide-et-impera* (branch-and-bound) fashion within a binary-tree search scheme.

Let us consider the case of $k$-HC$_{(2,1)}$. We assume (as done by most of the state-of-the-art solvers such as Gurobi Gurobi Optimization, LLC (2026)), that polyhedral envelopes are employed. Under such an assumption, when considering the nonlinear constraints $||w_j||_2^2 = \sum_{h=1}^n w_{jh}^2 \geq 1, j \in [k]$,

a classical SBB method first introduces the auxiliary variable $z_{jh}$ for each nonlinear term $w_{jh}^2$ and a corresponding defining constraint $z_{jh} = w_{jh}^2$. It then substitutes the original nonlinear constraint with $\sum_{h=1}^{n} z_{jh} \geq 1$. Each defining constraint is then relaxed into a polyhedral envelope. The point-wise minimal outer envelope of a bilinear product corresponds to the well-known McCormick envelope McCormick (1976).

## 4.2 BASELINE MATHEMATICAL PROGRAMMING FORMULATION FOR $k$-HC$_{(2,1)}$

We start by considering as baseline the following classical Mixed Integer Quadratically Constrained Quadratic Programming (MI-QCQP) formulation for $k$-HC$_{(2,1)}$ Coniglio (2011); Amaldi & Coniglio (2013):

$$(k\text{-HC}_{(2,1)}) \quad \min_{(w,\gamma),x,d} \left\{ \sum_{i=1}^{m} d_i^2 : \begin{array}{ll} \sum_{j=1}^{k} x_{ij} = 1 & \forall i \in [m] \\ \|w_j\|_2 \geq 1 & \forall j \in [k] \\ d_i \geq w_j^T a_i - \gamma_j - d^U(1 - x_{ij}) & \forall i \in [m], j \in [k] \\ d_i \geq -w_j^T a_i + \gamma_j - d^U(1 - x_{ij}) & \forall i \in [m], j \in [k] \end{array} \right\}.$$

In it, $x_{ij} \in \{0, 1\}$ takes value 1 if and only if $a_i$ is assigned to the hyperplane of index $j \in [k]$; $d_i$ is the distance between $a_i$ and the hyperplane of index $j \in [k]$; $d^U$ is an upper bound on the largest distance between any point $a_i$ and hyperplane of index $j \in [k]$. The only nonconvexity of the formulation is due to the 2-norm constraints. W.l.o.g., we assume $a_i \geq 0$ for all $i \in [m]$ (as this can be easily obtained in a preprocessing step by translating the dataset). The following bounds on the variables can be included. We let $d^U := \|b\,e\|_2$, where $e$ is the all-one vector and $b$ is the length of the edge of the smallest hypercube that contains $\{a_1, \ldots, a_m\}$. Since $\|w_j\|_2 = 1$ holds in any optimal solution and $\max\{\|w_j\|_\infty : \|w_j\|_2 = 1\} = 1$, we impose $\|w_j\|_\infty \leq 1$ via $-e \leq w_j \leq e$, $j \in [k]$. These bounds imply $-nb - d^U \leq \gamma_j \leq nb + d^U$, $j \in [k]$.

Since the point-to-hyperplane distance is symmetric, given any solution to $k$-HC$_{(2,1)}$, an equivalent one can be obtained by changing the sign of $w_j$ for some $j \in [k]$. To remove such a symmetry (symmetries are known to be a hindrance when solving mathematical programming problems to optimality via methods based on (spatial) branch-and-bound), we impose $w_j$ to belong to an arbitrary half-space of $\mathbb{R}^n$ for each $j \in [k]$ by imposing $w_{j1} \geq 0, j \in [k]$, where $w_{j1}$ is the first component of $w_j$. In this way, any solution that is obtainable by changing the sign of a component of one of the vectors $w_j$ becomes infeasible (due to being obtained from the previous one by reflection of $w_j$ over the hyperplane defining the halfspace that we selected), thus breaking the symmetry. In all our formulations, we partially remove the symmetry on $x_{ij}, i \in [m], j \in [k]$, that is induced by the assignment constraints by imposing $x_{ij} = 0$ for all $i, j \in [m] \times [k]$ with $i < j$. This reduces the number of 0-1 variables by $\frac{(k-1)k}{2}$.

## 4.3 SOLVING THE FORMULATION $\left(k\text{-HC}_{(2,1)}\right)$ VIA SBB

Let us now analyze the behavior of a standard SBB method employed for solving the classical formulation $(k\text{-HC}_{(2,1)})$. Since the projection onto the $w$ space of the feasible region of $k$-HC$_{(2,1)}$ is nonconvex and its complement is symmetric about the origin, any SBB method based on convex envelopes will necessarily convexify the infeasible region, thus making the trivial solution $w_j = 0, j \in [k]$, feasible. This leads to a bound as weak as possible due to the fact that the objective function is the sum of squares $\sum_{i=1}^{m} d_i^2 \geq 0$ and, with $(w_j, \gamma_j) = 0, j \in [k]$, we obtain $\sum_{i=1}^{m} d_i^2 = 0$.

The following assumption holds in most SBB codes—see, e.g., Belotti et al. (2009):

**Assumption 1.** *Assume that, when spatially branching on variables with a symmetric domain, branching takes place on the midpoint of the domain.*

Notice that, with the bounds we included, the domain of $w_{jh}, j \in [k], h \in [n]$, is symmetric.

Crucially, under Assumption 1 the geometry of the feasible region of $k$-HC$_{(2,1)}$ makes it so that the number of branching operations that are needed to make the 0 solution infeasible (and, thus, compute a nonzero global lower bound) is exponentially large (the proof is in the appendix):

**Proposition 2.** *Under Assumption 1, when solving $k$-HC$_{(2,1)}$ a nonzero lower bound is obtained only after generating at least $2^{k(n-1)}$ branching nodes.*

This is particularly bad since, until the first nonzero lower bound has been calculated, no pruning can happen on the tree due to the fact that a lower bound of 0 trivially holds at any node since the objective function is a sum of squares.

## 4.4 STRENGTHENED FORMULATIONS

We now construct valid formulations for $k$-HC$_2$ which are strengthened by featuring not only the 2-norm constraints but also a collection of polyhedral-norm constraints. Building on the relaxations we constructed before, we introduce the following three strengthened formulations (in each of them, the norm constraints are imposed for all $j \in [k]$):

$$(k\text{-HC}_{(2,1),(\infty,1)}) \qquad \min_{(w,\gamma)} \left\{ \sum_{i=1}^{m} \min_{j \in [k]} \left\{ (a_i^\top w_j - \gamma_j)^2 \right\} : \begin{array}{l} \|w_j\|_2 \geq 1 \\ \|w_j\|_1 \geq 1 \end{array} \right\}$$

$$(k\text{-HC}_{(2,1),(1,\frac{1}{\sqrt{n}})}) \qquad \min_{(w,\gamma)} \left\{ \sum_{i=1}^{m} \min_{j \in [k]} \left\{ (a_i^\top w_j - \gamma_j)^2 \right\} : \begin{array}{l} \|w_j\|_2 \geq 1 \\ \|w_j\|_\infty \geq \frac{1}{\sqrt{n}} \end{array} \right\}$$

$$(k\text{-HC}_{(2,1),(\text{multi},1)}) \qquad \min_{(w,\gamma)} \left\{ \sum_{i=1}^{m} \min_{j \in [k]} \left\{ (a_i^\top w_j - \gamma_j)^2 \right\} : \begin{array}{l} \|w_j\|_2 \geq 1 \\ \|w_j\|_1 \geq 1 \\ \|w_j\|_\infty \geq \frac{1}{\sqrt{n}} \end{array} \right\}.$$

Before analyzing the number of branching operations needed to achieve a nonzero lower bound with these formulations, we report the Mixed Integer Linear Programming (MILP) formulations by which we formulate the polyhedral-norm constraints.

**1-norm.** We formulate the constraints $\|w_j\|_1 \geq 1$, $j \in [k]$, via the following absolute-value reformulation:

$$w_{jh}^+ - w_{jh}^- = w_{jh} \qquad\qquad j \in [k], h \in [n] \tag{1a}$$

$$w_{jh}^+ \leq s_{jh} \qquad\qquad j \in [k], h \in [n] \tag{1b}$$

$$w_{jh}^- \leq (1 - s_{jh}) \qquad\qquad j \in [k], h \in [n] \tag{1c}$$

$$\sum_{h=1}^{n} (w_{jh}^+ + w_{jh}^-) \geq 1 \qquad\qquad j \in [k] \tag{1d}$$

$$0 \leq w_{jh}^+, w_{jh}^- \leq 1 \qquad\qquad j \in [k], h \in [n] \tag{1e}$$

$$s_{jh} \in \{0, 1\} \qquad\qquad j \in [k], h \in [n]. \tag{1f}$$

The binary variable $s_{jh}$ denotes the sign of the $h$-th component of $w_j$. Consider a component $w_{jh}$ of index $h$ of $w_j$. Due to Constraints (1a)–(1c), if $w_{jh} > 0$, then $w_{jh}^+ > 0$ (with $w_{jh}^+ = w_{jh}$ and $w_{jh}^- = 0$) and $s_{jh} = 1$. Otherwise, if $w_{jh} < 0$, then $w_{jh}^- > 0$ (with $w_{jh}^+ = 0$ and $w_{jh}^- = -w_{jh}$) and $s_{jh} = 0$. Since $w_j^+$ and $w_j^-$ are component-wise complementary thanks to Constraints (1b)–(1c), we deduce that $w_j^+ + w_j^- = |w_j|$ holds. Thus, Constraint (1d) guarantees $\|w_j\|_1 \geq 1$. When these constraints are imposed, we break symmetry as mentioned before by imposing $w_{j1} \geq 0$, $j \in [k]$. This leads to $s_{j1} = 1$ and $w_{j1}^- = 0$, thanks to which Constraint (1d) becomes $w_{j1} + \sum_{h=2}^{n}(w_{jh}^+ + w_{jh}^-) \geq 1$.

**$\infty$-norm.** We formulate the constraints $\|w_j\|_\infty \geq \frac{1}{\sqrt{n}}$, $j \in [k]$, i.e., $\max_{h \in [n]}\{|w_{jh}|\} \geq \frac{1}{\sqrt{n}}$, $j \in [k]$, as the disjunction $\bigvee_{h=1}^{n} \left( w_{jh} \leq -\frac{1}{\sqrt{n}} \vee w_{jh} \geq \frac{1}{\sqrt{n}} \right), j \in [k]$. Differently from the previous cases, in this case we break symmetry by (w.l.o.g.) always selecting $w_{jh} \geq \frac{1}{\sqrt{n}}$ from each elementary disjunction $w_{jh} \leq -\frac{1}{\sqrt{n}} \vee w_{jh} \geq \frac{1}{\sqrt{n}}$. This translates into considering the restricted disjunction $\bigvee_{h=1}^{n} \left( w_{jh} \geq \frac{1}{\sqrt{n}} \right), j \in [k]$. For each $j \in [k]$, we restate the resulting disjunctive set

via the following MILP formulation:

$$w_{jh} \geq -1 + \left(1 + \frac{1}{\sqrt{n}}\right) u_{jh} \qquad\qquad j \in [k], h \in [n] \qquad\qquad (2a)$$

$$\sum_{h=1}^{n} u_{jh} = 1 \qquad\qquad j \in [k] \qquad\qquad (2b)$$

$$u_{jh} \in \{0,1\} \qquad\qquad j \in [k], h \in [n]. \qquad\qquad (2c)$$

Due to Constraint (2a), if $u_{jh} = 1$ holds for some $h \in [n]$, then $w_{jh} \geq \frac{1}{\sqrt{n}}$ holds (the constraint is inactive if $u_{jh} = 0$, and reads $w_{jh} \geq -1$). Constraint (2b) imposes that exactly one component of $u_j = (u_{j1}, \ldots, u_{jn})$ is equal to 1.

When imposing multiple norm constraints at once, we only have to pay attention to the way symmetry is prevented, as the symmetry-breaking constraint $w_{j1} \geq 0$ we introduced for the constraints $\|w_j\|_2 \geq 1$, $j \in [k]$, and $\|w_j\|_1 \geq 1$, $j \in [k]$, is not compatible with the one-sided disjunction we considered for $\|w_j\|_\infty \geq \frac{1}{\sqrt{n}}$, $j \in [k]$, and imposing both would lead to an over-restriction. Whenever the $\|w_j\|_\infty \geq \frac{1}{\sqrt{n}}$ constraints are imposed, we resolve the issue by dropping the symmetry-breaking constraints $w_{j1} \geq 0$, $j \in [k]$.

### 4.5 SOLVING THE STRENGTHENED FORMULATIONS VIA SBB

We extend the analysis in Proposition 2 to the strengthened formulations with the following two propositions (the proofs of both are contained in the appendix):

**Proposition 3.** *Assume that the constraint $\|w_j\|_1 \geq 1$, $j \in [k]$, is imposed and that branching takes place on the $s_{jh}$ variables first. Then, a nonzero global lower bound is obtained after generating at least $2^{k(n-1)}$ nodes. If $k$-HC$_{(\infty,1)}$ is being solved, no further branching on $w$ takes place.*

**Proposition 4.** *Assume that $\|w_j\|_\infty \geq \frac{1}{\sqrt{n}}$, $j \in [k]$, is imposed and that branching takes place on the $u_{jh}$ variables first. Then, $k(n-1)$ nodes suffice to obtain a nonzero lower bound. If $k$-HC$_{(1,\frac{1}{\sqrt{n}})}$ is being solved, no further branching on $w$ takes place.*

Propositions 3 and 4 show the crucial advantages of strengthening formulation $(k$-HC$_{(2,1)})$ as we proposed via the two (scaled) polyhedral-norm constraints we considered. Proposition 3 indicates that, if the $\|w_j\|_1 \geq 1$, $j \in [k]$, constraints are imposed and branching takes place on the 0-1 variables of such norm constraints, in a complete SBB tree of depth $\Theta(2^{k(n-1)})$ the polyhedral-norm constraint is satisfied in *every* leaf node. This is in stark contrast to the 2-norm case, where the same number of branching operations only suffices to obtain the first nonzero global lower bound, and the number of branchings needed to completely describe the feasible region of the problem in the $w$ space depends on the solver's feasibility tolerance (since, for each $j \in [k]$, the complement of the feasible region is a sphere).

Crucially, Proposition 4 shows that, when the $\|w_j\|_\infty \geq \frac{1}{\sqrt{n}}$, $j \in [k]$, constraints are imposed and branching takes place on their 0-1 variables, the size of the SBB tree is extremely small—only polynomial in $k$ and $n$. The difference between the two results is due to the geometry of the 1- and $\infty$-norm balls with $n > 2$, since the former has $2^n$ facets while the latter only $2n$.

When included in a formulation for $k$-HC$_2$ on top of the constraints $\|w_j\|_2 \geq 1$, $j \in [k]$, the polyhedral-norm constraints substantially accelerate the computation of a nonzero global lower bound, leading to more pruning and, overall, to a faster SBB method. This is better shown in the next section.

## 5 COMPUTATIONAL RESULTS

We assess the effectiveness of our strengthened formulations with Gurobi 10's SBB using 12 threads on a 2.6GHz Intel Core i7-9750H equipped with 32 GB RAM, with a total time limit across the 12 cores of 168,000 seconds (46 hours).

We consider two testbeds: `Low-dim` and `High-dim`. `Low-dim` contains 43 instances with $m = 10, \ldots, 30$, $n = 2, 3$, and $k = 2, 3$. These instances are a superset of the 24 instances tackled with SBB techniques in Amaldi & Coniglio (2013). `High-dim` contains 43 instances with $m = 10, \ldots, 17$, $n = 2, 3, 4, 5$, and $k = 2, 3, 4, 5$. Both datasets are generated by randomly choosing $(w_j, \gamma_j)$, $j \in [k]$, with a uniform distribution in $[-1, 1]$ and distributing uniformly at random the $m$ points such that each of them belongs (with 0 distance) to a hyperplane. Then, an orthogonal deviation from the corresponding hyperplane is added to each point by sampling a Gaussian distribution with 0 mean and a variance that is selected, for each hyperplane, uniformly at random in $[0.7 \cdot 0.003, 0.003]$. Details on how to access and run our code as well as on how to access the dataset we used in the experiment are reported in the appendix.

We consider four formulations: $(k\text{-HC}_{(2,1)})$, $(k\text{-HC}_{(2,1),(1,\frac{1}{\sqrt{n}})})$, $(k\text{-HC}_{(2,1),(\infty,1)})$, and $(k\text{-HC}_{(2,1),(\text{multi},1)})$. Tables 1 and 2 report, for each formulation, the median computing time on the subset of instances solved by all four, the median speed-up relative to $(k\text{-HC}_{(2,1)})$, and the Holm-corrected (with a family-wise error rate $\alpha = 0.05$) $p$-value of a two-sided Wilcoxon signed-rank test against $(k\text{-HC}_{(2,1)})$.

Table 1: LowDim: comparison to $(k\text{-HC}_{(2,1)})$ on the 20 instances solved by all four formulations.

| Algorithm | Median (s) | Speed-up | $p$-value |
|---|---|---|---|
| $(k\text{-HC}_{(2,1)})$ | 169.9 | 1× | – |
| $(k\text{-HC}_{(2,1),(1,\frac{1}{\sqrt{n}})})$ | 4.15 | 40.9× | $1.1 \times 10^{-4}$ |
| $(k\text{-HC}_{(2,1),(\infty,1)})$ | 6.10 | 27.9× | $1.1 \times 10^{-4}$ |
| $(k\text{-HC}_{(\text{multi},1)})$ | 5.00 | 34.0× | $1.1 \times 10^{-4}$ |

Table 2: HighDim: comparison to $(k\text{-HC}_{(2,1)})$ on the 30 instances solved by all four formulations.

| Algorithm | Median (s) | Speed-up | $p$-value |
|---|---|---|---|
| $(k\text{-HC}_{(2,1)})$ | 208.6 | 1× | – |
| $(k\text{-HC}_{(2,1),(1,\frac{1}{\sqrt{n}})})$ | 18.20 | 11.5× | $5.6 \times 10^{-9}$ |
| $(k\text{-HC}_{(2,1),(\infty,1)})$ | 20.65 | 10.1× | $7.5 \times 10^{-9}$ |
| $(k\text{-HC}_{(\text{multi},1)})$ | 37.35 | 5.6× | $8.7 \times 10^{-4}$ |

Detailed results are reported in Tables 3 and 4. Let us focus first on the `Low-dim` testbed. With the three strengthened formulations $(k\text{-HC}_{(2,1),(1,\frac{1}{\sqrt{n}})})$, $(k\text{-HC}_{(2,1),(\infty,1)})$, and $(k\text{-HC}_{(2,1),(\text{multi},1)})$, 21 instances that are not solved in over 46 hours with the classical formulation $(k\text{-HC}_{(2,1)})$ are solved in under 2 hours. With the strengthened formulations, the 20 instances that are also solved with the classical formulation are solved, respectively, 41, 28, and 34 times faster. Incidentally, our results on the `Low-dim` testbed prove that all the heuristic solutions found in Amaldi & Coniglio (2013) on the 24 instances considered in that work (those with $m = 10, 14, 18, 22, 26, 30$) are optimal.

Let us turn now to the `High-dim` testbed. On it, with the best-performing strengthened formulation we manage to solve 10 more instances than with the classical formulation. With the strengthened formulations, the 30 instances that are also solved with the classical formulation are solved, respectively, 12, 10, and 6 times faster.

Notice that the speedup obtained with $(k\text{-HC}_{(2,1),(\text{multi},1)})$ is smaller than those obtained with $(k\text{-HC}_{(2,1),(\infty,1)})$ and $(k\text{-HC}_{(2,1),(1,\frac{1}{\sqrt{n}})})$. Such a behavior is well explained by the results of Propositions 3 and 4: As $n$ and $k$ increase, the difference between the exponential lower bound on the number of nodes required to obtain a nonzero global lower bound in the first proposition and the polynomial one in the second one becomes larger and larger. Thus, any branching operations taking place on the constraints $\|w_j\|_1 \geq 1$ have a much smaller impact on the bound than those taking place on the $\|w_j\|_\infty \geq \frac{1}{\sqrt{n}}$, $j \in [k]$, which explains the superior performance of $(k\text{-HC}_{(2,1),(1,\frac{1}{\sqrt{n}})})$.

## 6 CONCLUDING REMARKS

We have focused on solving the 2-norm $k$-Hyperplane Clustering problem with spatial branch-and-bound (SBB) techniques by strengthening the classical formulation with constraints that arise from (scaled) $p$-norm formulations of the problem, with $p \neq 2$. Focusing on the 1- and $\infty$-norms, we have theoretically shown that including the constraints stemming from the 1-norm version of the problem (featuring scaled $\infty$-norm constraints) leads to computing nonzero lower bounds in a linear (rather than exponential) number of SBB nodes. Our experimental results show very large speedups, reducing median solve times by up to 41× while increasing the total number of solved instances by up to 63%, substantially improving the problem's solvability to global optimality.

Table 3: Results on the `LowDim` dataset

| $m$ | $n$ | $k$ | $(k\text{-HC}_{(2,1)})$ time | obj | $(k\text{-HC}_{(2,1),(1,\cdot\frac{1}{\sqrt{n}})})$ time | obj | $(k\text{-HC}_{(2,1),(\infty,1)})$ time | obj | $(k\text{-HC}_{(multi,1)})$ time | obj |
|---|---|---|---|---|---|---|---|---|---|---|
| 10 | 2 | 2 | 0.3 | 0.3 | 0.2 | 0.3 | 0.2 | 0.3 | 0.2 | 0.3 |
| 10 | 2 | 3 | 0.7 | 0.5 | 1.0 | 0.5 | 0.8 | 0.5 | 1.0 | 0.5 |
| 14 | 2 | 2 | 1.6 | 8.5 | 0.6 | 8.5 | 0.2 | 8.5 | 0.3 | 8.5 |
| 14 | 2 | 3 | 31.9 | 0.8 | 4.4 | 0.8 | 3.4 | 0.8 | 5.4 | 0.8 |
| 18 | 2 | 2 | 13.9 | 3.4 | 0.4 | 3.4 | 0.4 | 3.4 | 0.7 | 3.4 |
| 18 | 2 | 3 | 488.9 | 0.7 | 3.9 | 0.7 | 4.4 | 0.7 | 4.6 | 0.7 |
| 22 | 2 | 2 | 179.2 | 9.7 | 1.7 | 9.7 | 1.4 | 9.7 | 0.9 | 9.7 |
| 22 | 2 | 3 | 2213.3 | 2.4 | 11.2 | 2.4 | 11.2 | 2.4 | 9.8 | 2.4 |
| 25 | 2 | 2 | 28.9 | 8.2 | 0.6 | 8.2 | 0.4 | 8.2 | 1.4 | 8.2 |
| 25 | 2 | 3 | 168000.0 | 2.7 | 936.6 | 2.7 | 96.1 | 2.7 | 221.0 | 2.7 |
| 26 | 2 | 2 | 168000.0 | – | 6.2 | 5.8 | 10.4 | 5.8 | 2.2 | 5.8 |
| 26 | 2 | 3 | 168000.0 | – | 39.2 | 3.4 | 56.6 | 3.4 | 28.3 | 3.4 |
| 27 | 2 | 2 | 168000.0 | – | 0.7 | 5.1 | 2.6 | 5.1 | 0.8 | 5.1 |
| 27 | 2 | 3 | 168000.0 | – | 1678.4 | 3.3 | 2687.7 | 3.3 | 238.6 | 3.3 |
| 28 | 2 | 2 | 168000.0 | – | 8.6 | 11.7 | 6.3 | 11.7 | 1.8 | 11.7 |
| 28 | 2 | 3 | 168000.0 | – | 293.1 | 3.6 | 471.3 | 3.6 | 153.5 | 3.6 |
| 29 | 2 | 2 | 168000.0 | – | 0.8 | 7.1 | 0.3 | 7.1 | 0.8 | 7.1 |
| 29 | 2 | 3 | 168000.0 | – | 7694.9 | 7.1 | 6029.0 | 7.1 | 1476.4 | 7.1 |
| 30 | 2 | 2 | 168000.0 | – | 10.4 | 9.1 | 38.5 | 9.1 | 1.6 | 9.1 |
| 30 | 2 | 3 | 168000.0 | – | 172.9 | 3.4 | 191.2 | 3.4 | 44.3 | 3.4 |
| 10 | 3 | 2 | 1.1 | 0.9 | 0.4 | 0.9 | 1.0 | 0.9 | 0.9 | 0.9 |
| 10 | 3 | 3 | 30.2 | 0.0 | 32.6 | 0.0 | 31.9 | 0.0 | 41.9 | 0.0 |
| 14 | 3 | 2 | 8.4 | 0.7 | 0.8 | 0.7 | 0.8 | 0.7 | 1.4 | 0.7 |
| 14 | 3 | 3 | 206.4 | 0.1 | 29.7 | 0.1 | 25.5 | 0.1 | 49.7 | 0.1 |
| 18 | 3 | 2 | 160.6 | 0.7 | 3.7 | 0.7 | 7.8 | 0.7 | 4.5 | 0.7 |
| 18 | 3 | 3 | 2234.9 | 0.4 | 93.4 | 0.4 | 91.6 | 0.4 | 157.9 | 0.4 |
| 22 | 3 | 2 | 625.0 | 4.3 | 15.6 | 4.3 | 11.3 | 4.3 | 10.8 | 4.3 |
| 22 | 3 | 3 | 135362.9 | 1.3 | 1089.5 | 1.3 | 638.2 | 1.3 | 1243.7 | 1.3 |
| 23 | 3 | 2 | 6459.4 | 0.9 | 8.1 | 0.9 | 45.5 | 0.9 | 10.1 | 0.9 |
| 24 | 3 | 2 | 18049.6 | 6.9 | 66.3 | 6.9 | 474.7 | 6.9 | 34.5 | 6.9 |
| 24 | 3 | 3 | 168000.0 | 1.7 | 2470.6 | 1.5 | 2716.7 | 1.5 | 3817.0 | 1.5 |
| 25 | 3 | 2 | 22886.9 | 5.7 | 70.7 | 5.7 | 28.1 | 5.7 | 14.2 | 5.7 |
| 25 | 3 | 3 | 168000.0 | 1.3 | 1952.3 | 1.3 | 5060.3 | 1.3 | 2885.1 | 1.3 |
| 26 | 3 | 2 | 168000.0 | – | 6.3 | 4.5 | 4.7 | 4.5 | 4.4 | 4.5 |
| 26 | 3 | 3 | 168000.0 | – | 5937.9 | 1.3 | 4345.7 | 1.3 | 2300.2 | 1.3 |
| 27 | 3 | 2 | 168000.0 | – | 215.1 | 3.4 | 1274.8 | 3.4 | 58.5 | 3.4 |
| 27 | 3 | 3 | 168000.0 | – | 52548.9 | 2.9 | 65949.3 | 2.9 | 35206.1 | 2.9 |
| 28 | 3 | 2 | 168000.0 | – | 31.1 | 3.6 | 1.7 | 3.6 | 10.2 | 3.6 |
| 28 | 3 | 3 | 168000.0 | – | 4234.9 | 1.4 | 74560.6 | 1.4 | 4180.9 | 1.4 |
| 29 | 3 | 2 | 168000.0 | – | 143.5 | 8.1 | 34.0 | 8.1 | 12.5 | 8.1 |
| 29 | 3 | 3 | 168000.0 | – | 168000.0 | 4.9 | 168000.0 | 4.9 | 168000.0 | 4.9 |
| 30 | 3 | 2 | 168000.0 | – | 8083.1 | 2.5 | 168000.0 | 2.5 | 3014.8 | 2.5 |
| 30 | 3 | 3 | 168000.0 | – | 23488.8 | 3.2 | 168000.0 | 3.2 | 6541.5 | 3.2 |
| **# Sol** | | | 20 | | 42 | | 40 | | 42 | |

Table 4: Results on the `HighDim` dataset

| $m$ | $n$ | $k$ | $(k\text{-HC}_{(2,1)})$ time | obj | $(k\text{-HC}_{(2,1),(1,\cdot\frac{1}{\sqrt{n}})})$ time | obj | $(k\text{-HC}_{(2,1),(\infty,1)})$ time | obj | $(k\text{-HC}_{(multi,1)})$ time | obj |
|---|---|---|---|---|---|---|---|---|---|---|
| 10 | 2 | 4 | 8.3 | 0.0 | 2.4 | 0.0 | 1.8 | 0.0 | 6.8 | 0.0 |
| 10 | 4 | 2 | 4.9 | 0.0 | 0.8 | 0.0 | 6.1 | 0.0 | 3.9 | 0.0 |
| 11 | 2 | 4 | 21.9 | 0.1 | 9.8 | 0.1 | 5.9 | 0.1 | 17.7 | 0.1 |
| 11 | 2 | 5 | 1264.3 | 0.0 | 392.8 | 0.0 | 300.2 | 0.0 | 2689.7 | 0.0 |
| 11 | 4 | 2 | 5.4 | 0.0 | 1.6 | 0.0 | 1.6 | 0.0 | 2.1 | 0.0 |
| 12 | 2 | 4 | 79.4 | 0.1 | 17.0 | 0.1 | 8.1 | 0.1 | 30.5 | 0.1 |
| 12 | 2 | 5 | 425.6 | 0.0 | 160.4 | 0.0 | 56.8 | 0.0 | 282.8 | 0.0 |
| 12 | 4 | 2 | 17.3 | 0.1 | 1.2 | 0.1 | 7.7 | 0.1 | 10.1 | 0.1 |
| 12 | 5 | 2 | 29.3 | 0.0 | 14.4 | 0.0 | 16.4 | 0.0 | 26.1 | 0.0 |
| 13 | 2 | 4 | 238.2 | 0.1 | 19.4 | 0.1 | 14.6 | 0.1 | 38.4 | 0.1 |
| 13 | 2 | 5 | 935.1 | 0.0 | 127.1 | 0.0 | 55.8 | 0.0 | 170.7 | 0.0 |
| 13 | 3 | 4 | 4143.7 | 0.0 | 7567.6 | 0.0 | 168000.0 | – | 168000.0 | – |
| 13 | 4 | 2 | 13.0 | 0.1 | 6.5 | 0.1 | 2.1 | 0.1 | 9.3 | 0.1 |
| 13 | 4 | 3 | 948.7 | 0.0 | 567.1 | 0.0 | 712.6 | 0.0 | 4625.7 | 0.0 |
| 13 | 5 | 2 | 47.0 | 0.1 | 11.1 | 0.1 | 19.8 | 0.1 | 28.3 | 0.1 |
| 14 | 2 | 4 | 683.1 | 0.2 | 22.4 | 0.2 | 12.2 | 0.2 | 55.8 | 0.2 |
| 14 | 2 | 5 | 6526.6 | 0.0 | 628.6 | 0.0 | 211.9 | 0.0 | 586.0 | 0.0 |
| 14 | 3 | 4 | 168000.0 | – | 2757.6 | 0.0 | 2784.8 | 0.0 | 7540.2 | 0.0 |
| 14 | 4 | 2 | 58.5 | 0.5 | 2.2 | 0.5 | 7.0 | 0.5 | 9.6 | 0.5 |
| 14 | 4 | 3 | 1447.5 | 0.0 | 687.9 | 0.0 | 890.5 | 0.0 | 6906.7 | 0.0 |
| 14 | 5 | 2 | 120.1 | 0.1 | 13.8 | 0.1 | 21.5 | 0.1 | 36.3 | 0.1 |
| 15 | 2 | 4 | 1350.6 | 0.3 | 32.9 | 0.3 | 23.4 | 0.3 | 54.4 | 0.3 |
| 15 | 2 | 5 | 5854.2 | 0.0 | 320.5 | 0.0 | 92.9 | 0.0 | 445.3 | 0.0 |
| 15 | 3 | 4 | 168000.0 | – | 2760.8 | 0.0 | 1772.1 | 0.0 | 168000.0 | – |
| 15 | 4 | 2 | 37.5 | 0.6 | 5.8 | 0.6 | 8.4 | 0.6 | 9.2 | 0.6 |
| 15 | 4 | 3 | 3803.0 | 0.0 | 515.6 | 0.0 | 439.4 | 0.0 | 2208.8 | 0.0 |
| 15 | 5 | 2 | 98.1 | 0.1 | 13.5 | 0.1 | 40.7 | 0.1 | 35.0 | 0.1 |
| 16 | 2 | 4 | 5827.2 | 0.2 | 119.6 | 0.2 | 28.9 | 0.2 | 67.3 | 0.2 |
| 16 | 2 | 5 | 168000.0 | – | 582.6 | 0.0 | 346.6 | 0.0 | 781.9 | 0.0 |
| 16 | 3 | 4 | 168000.0 | – | 4586.5 | 0.0 | 2407.2 | 0.0 | 168000.0 | – |
| 16 | 3 | 5 | 168000.0 | – | 168000.0 | – | 168000.0 | – | 168000.0 | – |
| 16 | 4 | 2 | 179.0 | 1.1 | 12.9 | 1.1 | 15.0 | 1.1 | 12.1 | 1.1 |
| 16 | 4 | 3 | 5144.2 | 0.0 | 554.5 | 0.0 | 601.1 | 0.0 | 2507.3 | 0.0 |
| 16 | 5 | 2 | 444.9 | 0.8 | 28.5 | 0.8 | 43.2 | 0.8 | 60.8 | 0.8 |
| 17 | 2 | 4 | 168000.0 | 0.2 | 37.1 | 0.2 | 42.1 | 0.2 | 69.2 | 0.2 |
| 17 | 2 | 5 | 168000.0 | 0.1 | 1452.3 | 0.1 | 999.4 | 0.1 | 1517.1 | 0.1 |
| 17 | 3 | 4 | 168000.0 | – | 4970.5 | 0.0 | 2553.9 | 0.0 | 168000.0 | – |
| 17 | 3 | 5 | 168000.0 | – | 168000.0 | – | 168000.0 | – | 168000.0 | – |
| 17 | 4 | 2 | 175.7 | 0.5 | 9.8 | 0.5 | 10.6 | 0.5 | 9.8 | 0.5 |
| 17 | 4 | 3 | 168000.0 | – | 904.1 | 0.0 | 967.5 | 0.0 | 3679.0 | 0.0 |
| 17 | 4 | 4 | 168000.0 | – | 8218.2 | 0.0 | 8102.3 | 0.0 | 8104.9 | 0.0 |
| 17 | 5 | 2 | 1092.7 | 1.4 | 87.0 | 1.4 | 97.4 | 1.4 | 101.0 | 1.4 |
| 17 | 5 | 3 | 168000.0 | – | 8116.4 | 0.0 | 8082.4 | 0.0 | 7910.9 | 0.0 |
| **# Sol** | | | 31 | | 41 | | 40 | | 37 | |

An interesting research direction for future work is exploring the connection between $k$-HC and subspace clustering, in particular related to the recent literature on coresets for projective clustering and subspace approximation Rademacher et al. (2005); Sohler & Woodruff (2018); Eiben et al. (2021). These techniques allow us to construct small, weighted subsets of data that preserve the clustering cost within a $(1 + \varepsilon)$ factor. Integrating such coreset constructions with our exact SBB-based solver could yield a hybrid approach (approximate in data, but exact in optimization), combining scalability with provable global optimality guarantees.

## ACKNOWLEDGEMENT OF SUPPORT

The author's work was partially supported by the European Union under Next Generation EU — the Italian National Recovery and Resilience Plan (PNRR), PRIN 2022 PNRR (project code P20227CTY3, CUP D53D23018800001), project title "HEXAGON: Highly-specialized EXact Algorithms for Grid Operations at the National level".

REPRODUCIBILITY STATEMENT

The author provided all the necessary information to facilitate the reproducibility of the results. The code developed for this work is made available online (see Appendix) and freely distributed under the MIT license.[4]

ETHICS STATEMENT

All datasets employed in this work are publicly available for research and contain no personally identifiable information or harmful content. The methods introduced in this paper have a societal impact comparable to that of any other clustering algorithm.

LLM USAGE STATEMENT

All technical content presented in this paper is entirely the work of the author, with LLMs serving only as an editorial tool.

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

## A  CODE REPOSITORY AND LICENSING

The code used for the experiments is freely available under the MIT license (`https://choosealicense.com/licenses/mit/`) and is available at `https://github.com/stefanoconiglio/khc-multinorm`.

## B  FURTHER COMPUTATIONAL RESULTS

Table 5 reports the total node counts for the `HighDim` dataset. This provides a clearer picture of relative tree sizes and convergence behavior across formulations. The results confirm the theoretical analysis, with the $(k\text{-HC}_{(2,1)})$, $(k\text{-HC}_{(2,1),(1,\frac{1}{\sqrt{n}})})$, $(k\text{-HC}_{(2,1),(\infty,1)})$, and $(k\text{-HC}_{(\text{multi},1)})$ formulations generating, respectively, 7,987,723.07, 3,201,881.49, 2,741,496.67, and 4,632,264.51 nodes on average.

Table 5: HighDim instances: total SBB node counts by formulation.

| $m$ | $n$ | $k$ | ($k$-HC$_{(2,1)}$) | ($k$-HC$_{(2,1),(1,\frac{1}{\sqrt{n}})}$) | ($k$-HC$_{(2,1),(\infty,1)}$) | ($k$-HC$_{(\text{multi},1)}$) |
|---|---|---|---|---|---|---|
| 10 | 2 | 4 | 38 392 | 16 357 | 10 301 | 23 588 |
| 10 | 4 | 2 | 17 033 | 4 739 | 18 112 | 16 937 |
| 11 | 2 | 4 | 78 168 | 29 502 | 17 868 | 41 565 |
| 11 | 2 | 5 | 4 654 600 | 1 151 890 | 958 992 | 8 332 930 |
| 11 | 4 | 2 | 21 404 | 8 701 | 6 989 | 8 584 |
| 12 | 2 | 4 | 287 736 | 45 100 | 22 786 | 75 515 |
| 12 | 2 | 5 | 1 228 060 | 440 626 | 169 063 | 687 073 |
| 12 | 4 | 2 | 61 743 | 5 820 | 19 455 | 26 611 |
| 12 | 5 | 2 | 85 702 | 42 506 | 36 266 | 60 555 |
| 13 | 2 | 4 | 791 227 | 52 720 | 41 656 | 99 091 |
| 13 | 2 | 5 | 2 621 440 | 347 674 | 154 070 | 383 310 |
| 13 | 3 | 4 | 11 864 400 | 19 678 500 | 23 810 300 | 18 600 100 |
| 13 | 4 | 2 | 41 063 | 19 382 | 9 097 | 24 744 |
| 13 | 4 | 3 | 2 629 260 | 1 284 080 | 1 646 530 | 9 724 370 |
| 13 | 5 | 2 | 139 309 | 24 034 | 40 346 | 48 397 |
| 14 | 2 | 4 | 2 355 780 | 61 667 | 34 681 | 166 633 |
| 14 | 2 | 5 | 19 826 000 | 2 014 470 | 715 573 | 1 582 970 |
| 14 | 3 | 4 | 21 011 800 | 5 928 130 | 6 446 500 | 16 330 600 |
| 14 | 4 | 2 | 197 555 | 10 637 | 15 814 | 19 679 |
| 14 | 4 | 3 | 3 653 010 | 1 509 690 | 1 914 490 | 14 396 900 |
| 14 | 5 | 2 | 367 215 | 34 060 | 40 631 | 67 349 |
| 15 | 2 | 4 | 4 819 300 | 88 293 | 63 762 | 142 499 |
| 15 | 2 | 5 | 15 129 700 | 815 961 | 240 698 | 1 057 150 |
| 15 | 3 | 4 | 20 797 200 | 6 821 170 | 4 221 290 | 17 132 800 |
| 15 | 4 | 2 | 123 055 | 14 603 | 21 152 | 19 678 |
| 15 | 4 | 3 | 9 399 350 | 1 068 560 | 949 518 | 4 432 280 |
| 15 | 5 | 2 | 285 279 | 25 590 | 83 182 | 63 958 |
| 16 | 2 | 4 | 20 072 700 | 387 933 | 86 122 | 177 715 |
| 16 | 2 | 5 | 18 348 500 | 1 839 480 | 977 328 | 1 834 550 |
| 16 | 3 | 4 | 18 615 700 | 12 285 100 | 5 353 010 | 16 241 100 |
| 16 | 3 | 5 | 16 743 900 | 16 840 700 | 15 925 600 | 14 859 600 |
| 16 | 4 | 2 | 622 968 | 29 784 | 34 762 | 25 049 |
| 16 | 4 | 3 | 12 307 300 | 1 297 660 | 1 169 750 | 4 721 850 |
| 16 | 5 | 2 | 1 411 490 | 66 069 | 100 491 | 107 740 |
| 17 | 2 | 4 | 23 783 300 | 108 023 | 120 569 | 176 621 |
| 17 | 2 | 5 | 18 531 600 | 4 274 570 | 3 036 550 | 3 629 780 |
| 17 | 3 | 4 | 18 137 400 | 10 911 800 | 5 561 730 | 16 517 600 |
| 17 | 3 | 5 | 16 997 500 | 15 721 000 | 16 018 000 | 13 877 100 |
| 17 | 4 | 2 | 599 093 | 22 341 | 26 660 | 18 985 |
| 17 | 4 | 3 | 18 403 800 | 2 103 800 | 2 034 520 | 7 360 030 |
| 17 | 4 | 4 | 16 580 800 | 14 203 900 | 13 111 500 | 12 754 700 |
| 17 | 5 | 2 | 3 190 660 | 187 282 | 217 843 | 189 688 |
| 17 | 5 | 3 | 16 600 600 | 15 857 000 | 12 400 800 | 13 129 400 |

## C  LIST OF OUR THEORETICAL RESULTS WITH THE CORRESPONDING PROOFS

**Proposition 1.** *Given a hyperplane $H := \{x \in \mathbb{R}^n : x^\top w = \gamma\}$ and a point $a \in \mathbb{R}^n$, the function $d_p(a, H) = \frac{|w^\top a - \gamma|}{\|w\|_{p'}}$, where $\frac{1}{p} + \frac{1}{p'} = 1$, is a nonconvex function of $(w, \gamma)$ for every $p \in \mathbb{N} \cup \{\infty\}$.*

*Proof.* By definition, $\frac{|w^\top a - \gamma|}{\|w\|_{p'}}$ is a convex function of $(w, \gamma)$ if and only if the following holds for every $(w_1, \gamma_1)$ and $(w_2, \gamma_2) \in \mathbb{R}^{n+1}$ and $\lambda \in [0, 1]$:

$$\lambda \frac{|w_1^\top a - \gamma_1|}{\|w_1\|_{p'}} + (1 - \lambda) \frac{|w_2^\top a - \gamma_2|}{\|w_2\|_{p'}} \geq$$
$$\frac{|(\lambda w_1 + (1 - \lambda) w_2)^\top a - (\lambda \gamma_1 + (1 - \lambda)\gamma_2)|}{\|\lambda w_1 + (1 - \lambda) w_2\|_{p'}}. \tag{3}$$

Let $p' \in \mathbb{N}$. Let $a = (0, 0)$ and consider two hyperplanes of parameters $w_1 := (1, -\frac{1}{5}), \gamma_1 = 1$ and $w_2 := (-\frac{1}{5}, 1), \gamma_2 = 1$. Let $\gamma := \gamma_1 = \gamma_2$. Letting $\lambda = \frac{1}{2}$, Inequality (3) reads:

$$\frac{1}{2} \frac{1}{\sqrt[p']{1 + \left(\frac{1}{5}\right)^{p'}}} + \frac{1}{2} \frac{1}{\sqrt[p']{1 + \left(\frac{1}{5}\right)^{p'}}} \geq \frac{1}{\sqrt[p']{\left(\frac{2}{5}\right)^{p'} + \left(\frac{2}{5}\right)^{p'}}}, \tag{4}$$

or, equivalently:

$$\sqrt[p']{\left(\frac{2}{5}\right)^{p'} + \left(\frac{2}{5}\right)^{p'}} \geq \sqrt[p']{1 + \left(\frac{1}{5}\right)^{p'}}.$$

Taking both sides to the $p'$-th power, we have $2\left(\frac{2}{5}\right)^{p'} \geq 1 + \left(\frac{1}{5}\right)^{p'}$. After moving 1 to the left-hand side and multiplying both sides by $5^{p'}$, we deduce $2 \cdot 2^{p'} - 1 \geq 5^{p'}$, which implies $2 \cdot 2^{p'} >$

$2 \cdot 2^{p'} - 1 \geq 5^{p'}$. As $\left(\frac{5}{2}\right)^{p'} > 2$ holds for every $p' \in \mathbb{N} \cup \{\infty\}$ (as one can see by setting $p'$ to its smallest value, i.e., setting $p' := 1$), Inequality (4) is proven not to hold for any choice of $p' \in \mathbb{N}$ ($p \in \mathbb{N} \setminus \{1\} \cup \{\infty\}$).

Let us consider the case $p' = \infty$ now. With $w_1 = (1, -\frac{1}{5})$ and $w_2 = (-\frac{1}{5}, 1)$, we have $\|w_1\|_\infty = \|w_2\|_\infty = 1$ and, with $\lambda = \frac{1}{2}$, we obtain $\left\|\frac{1}{2}(w_1 + w_2)\right\|_\infty = \|\frac{1}{2}(\frac{4}{5}, \frac{4}{5})\|_\infty = \frac{2}{5}$. Substituting these values directly into equation 3 leads to

$$\frac{1}{2} + \frac{1}{2} \geq \frac{5}{2},$$

which does not hold, showing that convexity fails to hold also for $p' = \infty$ ($p = 1$). $\qquad \square$

**Lemma 1.** *The solutions to $(k\text{-HC}_{(2,1)})$ and $(k\text{-HC}_2)$ coincide. Also, $(k\text{-HC}_{(p,c)})$ is quadratically homogeneous w.r.t. c, i.e., $\mathrm{OPT}(k\text{-HC}_{(p,c)}) = c^2 \, \mathrm{OPT}(k\text{-HC}_{(p,1)})$.*

*Proof.* We start by showing that $(k\text{-HC}_{(2,1)})$ and $(k\text{-HC}_2)$ are equivalent when $c = 1$ and $p = 2$. [5] Indeed, as $n$ points in general position fix a hyperplane in $\mathbb{R}^n$, only $n$ of the $n + 1$ parameters in $(w_j, \gamma_j) \in \mathbb{R}^{n+1}$ are independent. Thus, $\|w_j\|_2^2 = \|w_j\|_2 = 1$ can be imposed w.l.o.g. for all $j \in [k]$. Relaxing $\|w_j\|_2 = 1$ as $\|w_j\|_2 \geq 1$ is w.l.o.g. as the latter is tight in any optimal solution— indeed, if not, a strictly better solution can be found by scaling $(w_j, \gamma_j)$ by $\frac{1}{\|w_j\|_{p'}}$, $j \in [k]$. Let $\{(w_j, \gamma_j)\}_{j \in [k]}$ be an optimal solution to $(k\text{-HC}_{(p,c)})$. As argued, $\|w_j\|_{p'} = c$ holds. Let now $(w_j', \gamma_j') := \frac{(w_j, \gamma_j)}{c}$, $j \in [k]$. Such a scaled solution satisfies $\|w_j'\|_{p'} = 1$ for all $j \in [k]$ and, thus, is feasible for $(k\text{-HC}_{(p,1)})$. Its objective function value is $\frac{1}{c^2}$ times the one of $\{(w_j, \gamma_j)\}_{j \in [k]}$. Since such a multiplicative difference is a constant, the scaled solution is also optimal for $(k\text{-HC}_{(p,1)})$. Thus, we have $\mathrm{OPT}(k\text{-HC}_{(p,c)}) = c^2 \, \mathrm{OPT}(k\text{-HC}_{(p,1)})$. $\qquad \square$

**Theorem 1.** *Let $p, q \in \mathbb{N} \cup \{\infty\}$ and $c > 0$. The three positive scalars $\alpha(p,q), \beta(p,q), \delta(p,q)$ which, for all $x \in \mathbb{R}^n$, satisfy the congruence inequality $\alpha(p,q)\|x\|_p \leq \beta(p,q)\|x\|_q \leq \delta(p,q)\|x\|_p$ for $p, q \in \mathbb{N} \cup \{\infty\}$ also satisfy the optimal-value inequality $\frac{\alpha(p,q)^2}{\delta(p,q)^2} \, \mathrm{OPT}(k\text{-HC}_{(p,c)}) \leq \mathrm{OPT}\left(k\text{-HC}_{(q, c\frac{\beta(p,q)}{\delta(p,q)})}\right) \leq \mathrm{OPT}(k\text{-HC}_{(p,c)})$.*

*Proof.* The inequality

$$\min_{x \in X} f(x) \leq \min_{x \in X} f'(x) \leq \min_{x \in X} f''(x) \tag{5}$$

clearly holds for any three functions $f, f', f'' : X \to \mathbb{R}$ satisfying $f(x) \leq f'(x) \leq f''(x)$ for all $x \in X \subseteq \mathbb{R}^n$. Since vector norms in $\mathbb{R}^n$ are congruent, for every $p, q \in \mathbb{N} \cup \{\infty\}$ there are three positive scalars $\alpha(p,q), \beta(p,q), \delta(p,q)$ which satisfy $\alpha(p,q)\|x\|_p \leq \beta(p,q)\|x\|_q \leq \delta(p,q)\|x\|_p$ for $p, q \in \mathbb{N} \cup \{\infty\}$. Since, by definition, $d_p(a, H) = \min_{y \in H} \|a - y\|_p$, equation 5 leads to the following congruence relationship for point-to-hyperplane distances that holds for every hyperplane $H$ in $\mathbb{R}^n$ and point $a \in \mathbb{R}^n$:

$$\alpha(p,q) \, d_p(a, H) \leq \beta(p,q) d_q(a, H) \leq \delta(p,q) \, d_p(a, H). \tag{6}$$

Squaring equation 6 and letting $H_1, \ldots, H_k$ be an arbitrary choice of $k$ hyperplanes, another application of equation 5 leads to

$$\alpha(p,q)^2 \min_{j \in [k]} \{d^2(a_i, H_j)_p\} \leq \beta(p,q)^2 \min_{j \in [k]} \{d^2(a_i, H_j)_q\} \leq \delta(p,q)^2 \min_{j \in [k]} \{d^2(a_i, H_j)_p\}. \tag{7}$$

---

[5]This was already observed in Amaldi & Coniglio (2013). The proof we provide here will be useful in the following.

Summing equation 7 over all data points with unit multipliers, we obtain the following surrogate inequality:

$$\alpha(p,q)^2 \sum_{i=1}^{m} \min_{j \in [k]} \{d^2(a_i, H_j)_p\} \leq$$

$$\beta(p,q)^2 \sum_{i=1}^{m} \min_{j \in [k]} \{d^2(a_i, H_j)_q\} \leq$$

$$\delta(p,q)^2 \sum_{i=1}^{m} \min_{j \in [k]} \{d^2(a_i, H_j)_p\}.$$

Applying again equation 5 by letting the minimization consider the choice of optimal parameters for the hyperplanes $H_j, j \in [k]$, we deduce $\alpha(p,q)^2 \operatorname{OPT}(k\text{-HC}_{(p,1)}) \leq \beta(p,q)^2 \operatorname{OPT}(k\text{-HC}_{(q,1)}) \leq \delta(p,q)^2 \operatorname{OPT}(k\text{-HC}_{(p,1)})$. Multiplying through by $c^2$ and using Lemma 1, we obtain $\alpha(p,q)^2 \operatorname{OPT}(k\text{-HC}_{(p,c)}) \leq \beta(p,q)^2 \operatorname{OPT}(k\text{-HC}_{(q,c)}) \leq \delta(p,q)^2 \operatorname{OPT}(k\text{-HC}_{(p,c)})$. By using the quadratic homogeneity property of Lemma 1 one more time, we deduce $\beta(p,q)^2 \operatorname{OPT}(k\text{-HC}_{(q,c)}) = \operatorname{OPT}(k\text{-HC}_{(q,c\beta(p,q))})$, which allows us to write:

$$\alpha(p,q)^2 \operatorname{OPT}(k\text{-HC}_{(p,c)}) \leq \operatorname{OPT}(k\text{-HC}_{(q,c\beta(p,q))}) \leq \delta(p,q)^2 \operatorname{OPT}(k\text{-HC}_{(p,c)}).$$

Dividing all three terms by $\delta(p,q)^2$ and applying Lemma 1 one last time to remove the coefficient $\frac{1}{\delta(p,q)^2}$ that would otherwise multiply the inner term, the claim follows. $\square$

**Corollary 1.** $k$-HC$_{(\infty,1)}$ and $k$-HC$_{(1,\frac{1}{\sqrt{n}})}$ satisfy:

$$\frac{1}{n} \operatorname{OPT}(k\text{-HC}_{(2,1)}) \leq \operatorname{OPT}(k\text{-HC}_{(\infty,1)}) \leq \operatorname{OPT}(k\text{-HC}_{(2,1)})$$

$$\frac{1}{n} \operatorname{OPT}(k\text{-HC}_{(2,1)}) \leq \operatorname{OPT}(k\text{-HC}_{(1,\frac{1}{\sqrt{n}})}) \leq \operatorname{OPT}(k\text{-HC}_{(2,1)}).$$

*Proof.* We rely on the following congruence relationships (see Proposition 5 for their derivation):

$$\frac{1}{\sqrt{n}}\|x\|_2 \leq \|x\|_\infty \leq \|x\|_2 \qquad\qquad \frac{1}{\sqrt{n}}\|x\|_2 \leq \frac{1}{\sqrt{n}}\|x\|_1 \leq \|x\|_2.$$

Thanks to Theorem 1, $\frac{1}{\sqrt{n}}\|x\|_2 \leq \|x\|_\infty \leq \|x\|_2$ implies

$$\frac{1}{n} \operatorname{OPT}(k\text{-HC}_{(2,1)}) \leq \operatorname{OPT}(k\text{-HC}_{(\infty,1)}) \leq \operatorname{OPT}(k\text{-HC}_{(2,1)}).$$

Thanks to Theorem 1, $\frac{1}{\sqrt{n}}\|x\|_2 \leq \frac{1}{\sqrt{n}}\|x\|_1 \leq \|x\|_2$ implies

$$\frac{1}{n} \operatorname{OPT}(k\text{-HC}_{(2,1)}) \leq \frac{1}{n} \operatorname{OPT}(k\text{-HC}_{(1,1)}) \leq \operatorname{OPT}(k\text{-HC}_{(2,1)})$$

which, due to Lemma 1, implies

$$\frac{1}{n} \operatorname{OPT}(k\text{-HC}_{(2,1)}) \leq \operatorname{OPT}(k\text{-HC}_{(1,\frac{1}{\sqrt{n}})}) \leq \operatorname{OPT}(k\text{-HC}_{(2,1)}).$$

$\square$

**Lemma 2.** *Solving $k$-HC subject to $\min\{\|w\|_1, \sqrt{n}\|w\|_\infty\} \geq 1$ coincides with solving an unconstrained version of $k$-HC where the point-to-hyperplane distance between $a_i$ and $H_j$ is defined as $\max\{d_\infty(a_i, H_j), \frac{1}{\sqrt{n}}d_1(a_i, H_j)\}$.*

*Proof.* As a consequence of Lemma 1, imposing $\min\{\|w\|_1, \sqrt{n}\|w\|_\infty\} \geq 1$ in the context of $k$-HC implies imposing $\min\{\|w\|_1, \sqrt{n}\|w\|_\infty\} = 1$ in any optimal solution and, thus, accounting for the distance between $a_i$ and $H_j$ as $|a_i^\top w_j - \gamma| = \frac{|a_i^\top w_j - \gamma|}{\min\{\|w\|_1, \sqrt{n}\|w\|_\infty\}}$. We can rewrite the latter as $\max\{\frac{|a_i^\top w_j - \gamma|}{\|w\|_1}, \frac{|a_i^\top w_j - \gamma|}{\sqrt{n}\|w\|_\infty}\} = \max\{\frac{|a_i^\top w_j - \gamma|}{\|w\|_1}, \frac{1}{\sqrt{n}}\frac{|a_i^\top w_j - \gamma|}{\|w\|_\infty}\} = \max\{d_\infty(a_i, H_j), \frac{1}{\sqrt{n}}d_1(a_i, H_j)\}$. $\square$

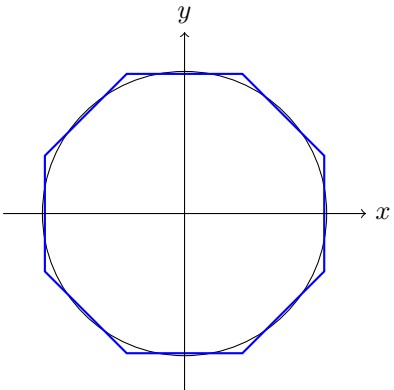

Figure 3: Sets of points satisfying $\|x\|_2 = 1$ (inner circle) and $\max\{\|x\|_\infty, \frac{1}{\sqrt{n}}\|x\|_1\} = 1$ (outer octagon).

**Lemma 3.** *The function* $\max\{d_\infty(a_i, H_j), \frac{1}{\sqrt{n}}d_1(a_i, H_j)\}$ *is a distance induced by the norm* $\max\{\|x\|_\infty, \frac{1}{\sqrt{n}}\|x\|_1\}$.

*Proof.* Let us show that $\max\{\|x\|_\infty, \frac{1}{\sqrt{n}}\|x\|_1\}$ is a norm in three steps.

*I. Positive definiteness.* First, it is clear that $\max\{\|x\|_\infty, \frac{1}{\sqrt{n}}\|x\|_1\} \geq 0$ and that $\max\{\|x\|_\infty, \frac{1}{\sqrt{n}}\|x\|_1\} = 0$ if and only if $x = 0$.

*II. Absolute homogeneity.* Second, it is also clear that $|\lambda|\max\{\|x\|_\infty, \frac{1}{\sqrt{n}}\|x\|_1\} = \max\{\lambda\|x\|_\infty, \lambda\frac{1}{\sqrt{n}}\|x\|_1\}$ for all $\lambda \in \mathbb{R}$.

*III. Triangle inequality.* Third, we must show that

$$\max\{\|x+y\|_\infty, \frac{1}{\sqrt{n}}\|x+y\|_1\} \leq \max\{\|x\|_\infty, \frac{1}{\sqrt{n}}\|x\|_1\} + \max\{\|y\|_\infty, \frac{1}{\sqrt{n}}\|y\|_1\} \quad (8)$$

holds for any $x, y \in \mathbb{R}^n$. To see this, we first notice that

$$\|x+y\|_\infty \leq \|x\|_\infty + \|y\|_\infty \qquad \text{and} \qquad \frac{1}{\sqrt{n}}\|x+y\|_1 \leq \frac{1}{\sqrt{n}}\|x\|_1 + \frac{1}{\sqrt{n}}\|y\|_1$$

hold since these functions are norms. Taking the maximum of the left-hand and right-hand sides of these two inequalities, thanks to the monotonicity of $\max$ we have:

$$\max\{\|x+y\|_\infty, \frac{1}{\sqrt{n}}\|x+y\|_1\} \leq \max\{\|x\|_\infty + \|y\|_\infty, \frac{1}{\sqrt{n}}\|x\|_1 + \frac{1}{\sqrt{n}}\|y\|_1\}. \quad (9)$$

To show that this implies that the triangle inequality is satisfied, we show that, for any $a, b, c, d \geq 0$, we have

$$\max\{a+c, b+d\} \leq \max\{a, b\} + \max\{c, d\}. \quad (10)$$

Trivially, we have $a \leq \max\{a,b\}$, $b \leq \max\{a,b\}$, $c \leq \max\{c,d\}$, and $d \leq \max\{c,d\}$. Adding the inequalities in pairs, we obtain $a+c \leq \max\{a,b\} + \max\{c,d\}$ and $b+d \leq \max\{a,b\} + \max\{c,d\}$. Taking the maximum of the left- and right-hand sides and applying again the monotonicity of $\max$, we deduce equation 10.

Letting now $a := \|x\|_\infty$, $c := \|y\|_\infty$, $b := \frac{1}{\sqrt{n}}\|x\|_1$, and $d := \frac{1}{\sqrt{n}}\|y\|_1$, from equation 10 we have

$$\max\{\|x\|_\infty + \|y\|_\infty, \frac{1}{\sqrt{n}}\|x\|_1 + \frac{1}{\sqrt{n}}\|y\|_1\} \leq \max\{\|x\|_\infty, \frac{1}{\sqrt{n}}\|x\|_1\} + \max\{\|y\|_\infty, \frac{1}{\sqrt{n}}\|y\|_1\}. \quad (11)$$

Combining equation 11 with equation 9, equation 8 is proven.

We have shown that $\max\{||x||_\infty, \frac{1}{\sqrt{n}}||x||_1\}$ is a norm. Showing that $\max\{d_\infty(a_i, H_j), \frac{1}{\sqrt{n}}d_1(a_i, H_j)\}$ is a distance follows straightforwardly by following the classical construction of point-to-hyperplane distances. $\square$

An illustration of the function $\max\{||x||_\infty, \frac{1}{\sqrt{n}}||x||_1\}$ is provided in Figure 3.

**Lemma 4.** *The norm* $\max\{||x||_\infty, \frac{1}{\sqrt{n}}||x||_1\}$ *satisfies the congruence inequality*

$$n^{-\frac{1}{4}}\|x\|_2 \le \max\Big\{ \|x\|_\infty, \ \frac{1}{\sqrt{n}}\|x\|_1 \Big\} \le \|x\|_2.$$

*Proof.* We prove the congruence relationship in two steps.

*I. Second part.* From the second part of each of the two congruence relationships

$$\frac{1}{\sqrt{n}}\|x\|_2 \le \|x\|_\infty \le \|x\|_2 \qquad\qquad \frac{1}{\sqrt{n}}\|x\|_2 \le \frac{1}{\sqrt{n}}\|x\|_1 \le \|x\|_2,$$

we directly deduce $\max\{||x||_\infty, \frac{1}{\sqrt{n}}||x||_1\} \le ||x||_2$.

*II. First part.* To prove the first part of the congruence, we establish what the largest value of $\|x\|_2$ is when $x$ is subject to $\max\{||x||_\infty, \frac{1}{\sqrt{n}}||x||_1\} \le 1$.

Let $S := \{x \in \mathbb{R}^n : \|x\|_\infty \le 1, \ \frac{1}{\sqrt{n}}\|x\|_1 \le 1\}$, or, equivalently, $S := \{x \in \mathbb{R}^n : \|x\|_\infty \le 1, \ \|x\|_1 \le \sqrt{n}\}$. Let $r$ be the fractional part of $\sqrt{n}$, i.e., $r := \sqrt{n} - \lfloor\sqrt{n}\rfloor \in [0,1)$. We'll prove that every maximizer of $\|x\|_2$ over $S$ has at most one fractional component in $(0,1)$ and, in particular, that $x^\star = (\underbrace{1,\dots,1}_{\lfloor\sqrt{n}\rfloor\text{ times}}, r, 0, \dots, 0)$ is one such maximizer with objective function value

$$\max_{x\in S}\|x\|_2 = \sqrt{\lfloor\sqrt{n}\rfloor + r^2}.$$

Since $S$ is symmetric under sign flips and coordinate permutations, we can w.l.o.g. restrict ourselves to vectors $x \in \mathbb{R}^n$ with $x_1 \ge x_2 \ge \cdots \ge x_n \ge 0$ and consider the equivalent problem

$$(P) \qquad \max\left\{\sum_{i=1}^n x_i^2 : \sum_{i=1}^n x_i \le \sqrt{n}, x \in [0,1]^n\right\}.$$

*(i)* First, we show that constraint $\sum_{i=1}^n x_i \le \sqrt{n}$ is tight in any optimal solution. This is because, if not, we could increase each $x_i$ until either $x_i = 1$ or $\sum_{i=1}^n x_i = \sqrt{n}$, thereby increasing the objective function $\sum_{i=1}^n x_i^2$.

*(ii)* Second, we show that any optimal solution features at most one fractional component. Suppose that $x$ is feasible with $\sum_{i=1}^n x_i = \sqrt{n}$ with $0 < x_i < 1$ and $0 < x_j < 1$ for some $i \neq j \in [n]$. W.l.o.g., assume $x_i \ge x_j$. Pick some $\varepsilon > 0$ with $x_i + \varepsilon \le 1$ and $x_j - \varepsilon \ge 0$, and define $\tilde{x}$ as $\tilde{x}_i := x_i + \varepsilon, \tilde{x}_j = x_j - \varepsilon$, and $\tilde{x}_k = x_k$ for all $k \in [n]$ with $k \neq i, j$. Then,

$$\sum_{i=1}^n \tilde{x}_i^2 - \sum_{i=1}^n x_i^2 = (x_i + \varepsilon)^2 + (x_j - \varepsilon)^2 - x_i^2 - x_j^2 = 2\varepsilon\underbrace{(x_i - x_j)}_{\ge 0} + 2\varepsilon^2 > 0.$$

This shows that any $x$ with two fractional entries is suboptimal.

*(iii)* Let $x$ be an optimal solution with $t$ ones, one fractional component $r \in [0, 1)$ (or none if $r = 0$), and $n - t - 1$ zeros. Since $\sum_{i=1}^n x_i \le \sqrt{n}$ is tight, we deduce $t + r = \sqrt{n}$, which (since $t$ is integer and $r < 1$) implies $t = \lfloor\sqrt{n}\rfloor$ and $r = \sqrt{n} - \lfloor\sqrt{n}\rfloor$. The objective value of $(P)$ is therefore $\sum_{i=1}^n x_i^2 = t\cdot 1^2 + r^2$, leading to $\|x\|_2 = \sqrt{\lfloor\sqrt{n}\rfloor + (\sqrt{n} - \lfloor\sqrt{n}\rfloor)^2}$. Since $0 \le \sqrt{n} - \lfloor\sqrt{n}\rfloor < 1$, we have $(\sqrt{n} - \lfloor\sqrt{n}\rfloor)^2 \le \sqrt{n} - \lfloor\sqrt{n}\rfloor$. Therefore, $\lfloor\sqrt{n}\rfloor + (\sqrt{n} - \lfloor\sqrt{n}\rfloor)^2 \le \lfloor\sqrt{n}\rfloor + \sqrt{n} - \lfloor\sqrt{n}\rfloor = \sqrt{n}$. Taking square roots gives $\sqrt{\lfloor\sqrt{n}\rfloor + (\sqrt{n} - \lfloor\sqrt{n}\rfloor)^2} \le \sqrt{\sqrt{n}} = n^{1/4}$.

*(iv)* With steps (i)–(iii), we have shown that, for every $x \in \mathbb{R}^n$ with $\max\{||x||_\infty, \frac{1}{\sqrt{n}}||x||_1\} \le 1$, we have $\|x\|_2 \le n^{1/4}$. For an arbitrary $x \neq 0$ (assuming this is w.l.o.g. since, for $x = 0$, the

congruence we are trying to prove is trivially satisfied), let $y := \frac{1}{\max\{||x||_\infty, \frac{1}{\sqrt{n}}||x||_1\}} x$. Clearly, $\max\{||y||_\infty, \frac{1}{\sqrt{n}}||y||_1\} = 1$. Thus, $y \in S$ and, thus, $||y||_2 \le n^{\frac{1}{4}}$. Therefore

$$||x||_2 = \max\{||x||_\infty, \frac{1}{\sqrt{n}}||x||_1\}||y||_2 \le \max\{||x||_\infty, \frac{1}{\sqrt{n}}||x||_1\}\, n^{\frac{1}{4}}.$$

It follows that

$$n^{-\frac{1}{4}}||x||_2 \le \max\{||x||_\infty, \frac{1}{\sqrt{n}}||x||_1\},$$

which concludes the proof. $\square$

**Corollary 2.** *Combining Lemma 4 with Theorem 1, the multi-norm relaxation $k$-HC$_{(\text{multi},1)}$ satisfies*

$$\frac{1}{\sqrt{n}}\,\text{OPT}\big(k\text{-HC}_{(2,1)}\big) \le \text{OPT}\big(k\text{-HC}_{(\text{multi},1)}\big) \le \text{OPT}\big(k\text{-HC}_{(2,1)}\big).$$

*Proof.* A direct consequence of applying Theorem 1 to the congruence relationship derived in Lemma 4. $\square$

**Proposition 2.** *Under Assumption 1, when solving $k$-HC$_{(2,1)}$ a nonzero lower bound is obtained only after generating at least $2^{k(n-1)}$ branching nodes.*

*Proof.* By assumption, each branching operation decides the sign of a component of $w_j$ for some $j \in [k]$ by splitting (with a half-space constraint) its feasible region with a hyperplane containing the origin. As long as the cone, call it $C$, obtained by intersecting such half-spaces is not pointed, the convex hull of its intersection with the feasible region of the problem contains the origin. Thus, the solution with $(w_j, \gamma_j) = 0$ and $x_{ij} = 1$, $i \in [m]$, which coincides with assigning every data point to the degenerate hyperplane of index $j$ (thus achieving $d_i = 0$, $i \in [m]$), is optimal regardless of the convex envelope that is employed. Only after branching has been carried out on each component of $w_j$ whose sign is not already restricted by the symmetry-breaking constraint (i.e., all coordinates except $w_{j1}$) for each $j \in [k]$, the cone $C$ becomes pointed and, thus, the convex hull of its intersection with the feasible region of the problem renders the trivial solution $(w_j, \gamma_j) = 0$, $j \in [k]$, infeasible, allowing for the calculation of a nonzero lower bound. This requires generating at least $2^{k(n-1)}$ nodes by branching on each hyperplane $n-1$ times (rather than $n$ due to the symmetry breaking constraint being already imposed). Notice that, in the general case, more branching operations are needed due to the $x$ variables being binary. $\square$

**Proposition 3.** *Assume that the constraint $\|w_j\|_1 \ge 1$, $j \in [k]$, is imposed and that branching takes place on the $s_{jh}$ variables first. Then, a nonzero global lower bound is obtained after generating at least $2^{k(n-1)}$ nodes. If $k$-HC$_{(\infty,1)}$ is being solved, no further branching on $w$ takes place.*

*Proof.* Let's consider a hyperplane of index $j \in [k]$. Let $s_{jh} = \frac{1}{2}$ for all $h \in [n]$, which implies $w_{jh}^+ \le \frac{1}{2}$ and $w_{jh}^- \le \frac{1}{2}$. Letting $w_{jh}^+ = w_{jh}^- = \frac{1}{2}$, we have $w_{jh}^+ + w_{jh}^- = 1$. This feasible solution trivially satisfies the 1-norm constraint equation 1d with $w_{jh}^+ - w_{jh}^- = w_{jh} = 0$. Thus, $(w_j, \gamma_j) = 0$, $j \in [k]$, is optimal. By branching on a variable $s_{jh}$, we impose either $w_{jh} \le 0$ (with $s_{jh} = 0$) or $w_{jh} \ge 0$ (with $s_{jh} = 1$). In both cases, the solution where $w_{jh}^+ = w_{jh}^- = \frac{1}{2}$ and $w_{jh} = 0$ becomes infeasible due to either $w_{jh}^+$ or $w_{jh}^-$ being forced to 0; the solution with $w_{jh'} = 0$, though, for any other $h' \in [n] \setminus \{h\}$, remains feasible as long as branching on it has not taken place. Thus, a nonzero lower bound is obtained after $2^{k(n-1)}$ branching nodes have been generated. If $k$-HC$_{(\infty,1)}$ is being solved, when such an exponentially-large tree of depth $k(n-1)$ is complete, though, $\|w_j\|_1 \ge 1$, $j \in [k]$, holds in each leaf node and, thus, no further branching on $w$ is necessary. $\square$

**Proposition 4.** *Assume that $\|w_j\|_\infty \ge \frac{1}{\sqrt{n}}$, $j \in [k]$, is imposed and that branching takes place on the $u_{jh}$ variables first. Then, $k(n-1)$ nodes suffice to obtain a nonzero lower bound. If $k$-HC$_{(1, \frac{1}{\sqrt{n}})}$ is being solved, no further branching on $w$ takes place.*

*Proof.* After branching on $u_{jh}$ for any pair $j, h$, the (left, w.l.o.g.) child node with $u_{jh} = 1$ satisfies $w_{jh} \geq \frac{1}{\sqrt{n}}$. This guarantees $||w_j||_\infty \geq \frac{1}{\sqrt{n}}$ and, thus, no further branching is needed on $w_j$ in the descendants of the left node. Further branching operations on $w_j$ are only necessary on the right child node where $u_{jh} = 0$ has been imposed. By iteratively applying this reasoning $n - 1$ times (recall that no disjunction is imposed on $w_{1j}$ due to symmetry breaking) for each $j \in [k]$, we obtain a tree with exactly two nodes per level (except for the root node) where each left node satisfies the $||w_j||_\infty \geq \frac{1}{\sqrt{n}}$ constraint for at least a $j \in [k]$. Therefore, when the tree has depth $k(n - 1)$, $||w_j||_\infty \geq \frac{1}{\sqrt{n}}$ is satisfied for all $j \in [k]$. When such an polynomially-sized tree of depth $k(n-1)$ is complete, $||w_j||_\infty \geq \frac{1}{\sqrt{n}}, j \in [k]$, holds in each leaf node and, thus, if $khcTwo1\frac{1}{\sqrt{n}}$ is being solved, no further branching on $w$ is necessary. $\qquad\square$

## D  PROOF OF THE APPROXIMATION FACTORS AND OF THEIR TIGHTNESS

We will rely on the following Lemma:

**Lemma 5.** *Given two functions $f, g : \mathbb{R}^n \to \mathbb{R}$ with $g$ surjective we have:*

$$\max_{x \in \mathbb{R}^n} \frac{f(x)}{g(x)} = \max_{\nu \in \mathbb{R}} \left\{ \max_{x \in \mathbb{R}^n} \left\{ \frac{f(x)}{\nu} : g(x) = \nu \right\} \right\}. \tag{12}$$

*If, for all $x \in \mathbb{R}^n$, $f(x) = f(|x|)$ and $g(x) = g(|x|)$, then:*

$$\max_{x \in \mathbb{R}^n} \frac{f(x)}{g(x)} = \max_{\nu \in \mathbb{R}_+} \left\{ \max_{x \in \mathbb{R}^n_+} \left\{ \frac{f(x)}{\nu} : g(x) = \nu \right\} \right\}. \tag{13}$$

*Proof.* If $g$ is surjective, then $\cup_{\nu \in \mathbb{R}} \{x \in \mathbb{R}^n : g(x) = \nu\} = \mathbb{R}^n$. We can therefore partition $\mathbb{R}^n$ into infinitely many subsets of type $\{x \in \mathbb{R}^n : g(x) = \nu\}$. An optimal solution to $\max_{x \in \mathbb{R}^n} \frac{f(x)}{g(x)}$ thus corresponds to the best solution over all such subsets. The special case in equation 13 follows by a similar argument. $\qquad\square$

**Proposition 5.** *The following relationships are satisfied for every $x \in \mathbb{R}^n$:*

$$||x||_2 \leq ||x||_1 \leq \sqrt{n}||x||_2$$

$$\frac{1}{\sqrt{n}}||x||_2 \leq ||x||_\infty \leq ||x||_2$$

*and the factors $\sqrt{n}$ and $\frac{1}{\sqrt{n}}$ are tight.*

*Proof.* We are looking for four positive coefficients $\alpha_1, \beta_1, \alpha_\infty, \beta_\infty$ that satisfy the following relationships for all $x \in \mathbb{R}^n$:

$$\alpha_1 ||x||_2 \leq ||x||_1 \leq \beta_1 ||x||_2$$
$$\alpha_\infty ||x||_2 \leq ||x||_\infty \leq \beta_\infty ||x||_2.$$

Assuming $x \neq 0$ as, for $x = 0$, $\alpha ||x||_p \leq ||x||_q \leq \beta ||x||_p$ holds for all $\alpha, \beta$ and for all $p, q \in \mathbb{N} \cup \{\infty\}$, the tightest values for $\alpha_1, \beta_1, \alpha_\infty, \beta_\infty$ must satisfy the following relationships:

$$\beta_1 = \max_{x \in \mathbb{R}^n} \frac{||x||_1}{||x||_2} \qquad\qquad \beta_\infty = \max_{x \in \mathbb{R}^n} \frac{||x||_\infty}{||x||_2}$$

$$\alpha_1 = \min_{x \in \mathbb{R}^n} \frac{||x||_1}{||x||_2} \qquad\qquad \alpha_\infty = \min_{x \in \mathbb{R}^n} \frac{||x||_\infty}{||x||_2}.$$

As it is not hard to see, $\max \frac{||x||_p}{||x||_q} = \min \frac{||x||_q}{||x||_p}$ holds for all $p, q \in \mathbb{N} \cup \{\infty\}$. Thus, we need to solve the following four problems:

$$\beta_1 = \max \frac{||x||_1}{||x||_2} \qquad\qquad \beta_\infty = \max \frac{||x||_\infty}{||x||_2}$$

$$\alpha_1 = \max \frac{||x||_2}{||x||_1} \qquad\qquad \alpha_\infty = \max \frac{||x||_2}{||x||_\infty}.$$

Let us consider the case of $\alpha_1, \alpha_\infty$, for which we are solving $\max \frac{\|x\|_2}{\|x\|_q}$ for $q = 1, \infty$. By virtue of Lemma 5, we are thus solving:

$$\alpha_q = \max_{\nu \in \mathbb{R}_+} \left\{ \frac{1}{\nu} \max_{x \in \mathbb{R}^n_+} \{\|x\|_2 : \|x\|_q = \nu\} \right\}.$$

As the maximum of a convex function (such as $\|x\|_2$) over a closed, convex set is achieved on the border of the latter and, if we are optimizing over a polytope, over its extreme vertices, we can w.l.o.g. relax $\|x\|_q = \nu$ into $\|x\|_q \le \nu$.

For $\alpha_1$, the extreme points of $\{x \in \mathbb{R}^n : \|x\|_1 \le \nu\}$ are of the form: $\nu e_\ell$ for all $\ell \in [n]$, with $e_\ell$ being the $\ell$-th canonical vector of $\mathbb{R}^n$. For each of them, we have $\|\nu e_\ell\|_2 = \sqrt{\nu^2} = \nu$. Thus, $\alpha_1 = \max \frac{\|x\|_2}{\|x\|_1} = \frac{\nu}{\nu} = 1$.

For $\alpha_\infty$, the extreme points of $\{x \in \mathbb{R}^n : \|x\|_\infty \le \nu\}$ are of the form: $(\pm\nu, \ldots, \pm\nu)$ for all possible choices of $\pm$. For each of them, we have $\|(\pm\nu, \ldots, \pm\nu)\|_2 = \sqrt{\nu^2 n} = \nu\sqrt{n}$. Thus, $\alpha_\infty = \max \frac{\|x\|_2}{\|x\|_\infty} = \frac{\nu\sqrt{n}}{\nu} = \sqrt{n}$.

Let us now consider the case of $\beta_1$ and $\beta_\infty$, for which we are solving $\max \frac{\|x\|_q}{\|x\|_2}$ for $q = 1, \infty$. By virtue of Lemma 5, we are thus solving:

$$\beta_q = \max_{\nu \in \mathbb{R}_+} \left\{ \frac{1}{\nu} \max_{x \in \mathbb{R}^n_+} \{\|x\|_q : \|x\|_2 = \nu\} \right\}.$$

For $\beta_1$, the problem reads:

$$\beta_1 = \max_{\nu \ge 0} \left\{ \frac{1}{\nu} \max_{x \in \mathbb{R}^n_+} \{e^T x : x^T x = \nu^2\} \right\}. \tag{14}$$

The KKT conditions for the relaxation of the inner problem of equation 14 obtained after dropping the nonnegativity on $x$ read:

$$\nabla_x (e^T x - \lambda(x^T x - \nu^2)) = 0$$
$$x^T x = \nu^2,$$

with $\lambda$ unrestricted in sign. From the first equation, we deduce $x = \frac{e}{2\lambda}$. By substituting it in the second equation, we obtain $\frac{e^T e}{2^2 \lambda^2} = \nu^2$, that is, $\lambda = \frac{\sqrt{n}}{2\nu}$. Thus, we have $x = \frac{e}{\sqrt{n}}\nu$. Since the latter is nonnegative, it is an optimal solution to both the relaxation of the inner problem of equation 14 with $x \in \mathbb{R}^n$ and its unrelaxed version with $x \in \mathbb{R}^n_+$. We thus have $\|x\|_1 = \frac{\nu}{\sqrt{n}}\|e\|_1 = \frac{\nu n}{\sqrt{n}} = \nu\sqrt{n}$. We conclude that $\beta_1 = \frac{\nu\sqrt{n}}{\nu} = \sqrt{n}$.

For $\beta_\infty$, the problem reads:

$$\beta_\infty = \max_{\nu \ge 0} \left\{ \frac{1}{\nu} \max_{x \in \mathbb{R}^n_+} \left\{ \max_{\ell \in [n]} \{x_\ell\} : x^T x = \nu^2 \right\} \right\}.$$

The optimal solutions to the inner problem are of the form $\nu e_\ell$, where $e_\ell$ is a canonical vector of $\mathbb{R}^n$, for which we have $\|\nu e_\ell\|_\infty = \nu$. We conclude that $\beta_\infty = \frac{\nu}{\nu} = 1$. $\quad\square$

