# OpenReview forum: "Solving the 2-norm k-hyperplane clustering problem via multi-norm formulations"
_ICLR.cc/2026/Conference — ICLR 2026 Poster_

### Official Review · Reviewer_RjaT · 2025-10-24

**Soundness:** 3
**Presentation:** 4
**Contribution:** 3
**Rating:** 8
**Confidence:** 3

**Summary:**

This paper focuses on improving spatial branch-and-bound (SBB) algorithms for optimally solving the 2-norm k-hyperplane clustering $k$-$HC_2$ problem, which seeks to choose k hyperplanes in a way that minimize the squared norm distance between each point and its nearest hyperplane.

The paper first considers a generalized objective $k$-$HC_{(p,c)}$ that considers the $p$-norm and a scaling constant $c$ in the constraints, and proves that optimal solution for  $k$-$HC_{(q,c')}$ for a careful choice of $c$ provides an optimal solution for $k$-$HC_{(p,c)}$. These bounds can be used specifically for the purpose of finding problems that approximate $k$-$HC_{(2,2)}$ = $k$-$HC_{2}$. The paper then focuses especially on approximations that are obtained for $k$-$HC_{2}$ in terms of $k$-$HC_{(\infty,1)}$ and $k$-$HC_{(1,1/\sqrt{n})}$. They then show how this leads to a strengthened formulation for  $k$-$HC_{2}$, which has a faster solution using SBB techniques. In particular, Section 4 proves that only a linear number of nodes in the branch-and-bound is needed to get a non-zero global lower bound for the new formulation, whereas an exponential number of nodes are needed to get a non-zero global lower bound for a basic formulation.

The paper shows in numerical experiments that the new approach leads to faster solve times in practice.

**Strengths:**

The topic of the paper is interesting and well-motivated from previous research.

The mathematical contributions of the paper is non-trivial; the technical contribution of the paper is quite high.

Even though the paper is proving a large number of non-trivial technical results, the presentation is very good and explains the main ideas and components (and how they fit together) very well. The approach makes sense at a high level, and the paper does a good job presenting enough technical detail in the main text while knowing what to push to the appendix.

The technical approach leads to concrete improvements in numerical experiments.

**Weaknesses:**

There were a couple places where the technical details were a little unclear and could have been explained a bit better:

Lemma 2 seems written somewhat informally and I do not follow what it is saying. In particular, it states that imposing a certain inequality "coincides with accounting for each point to hyperplane distance as ...". I don't understand precise meaning of the wording "coincides with accounting for", and hence I do not follow the meaning or significance of this lemma.

It's also not that clear how Lemma 2 is used in the main results. It's clear how Lemma 3 and Theorem 2 together produce the useful bound in corollary 2, but what role does Lemma 2 play in all this?

Figures 1 and 2 could use a bit more explanation, e.g., in the caption. At first something seemed backwards, because the text explained how the feasible region of a (q,c) problem with $q = 1, \infty$ and $c = 1, 1/\sqrt{n}$ contains the feasible region for $p = 2, c = 1$, but this appeared opposite to the pictures where a circle contained a red diamond and green square. It did not take me long to realize that this is because the feasible region is everything that lies *outside* these colored shapes, but it would have made things a lot clearer a lot quicker to note this explicitly, rather than just stating these are illustrations of certain feasible regions. Also, in the captions, shouldn't you write 2 instead of n in $\mathbb{R}^n$?

This is minor, but there are also some typos to fix:

* lemma 1: coincide Also ---> add missing period
* line 072: "d_p intrinsically" --> missing the word "is"?
* line 055: "nonzero global lower bounds is" --> bound is or bounds are

**Questions:**

Can you clarify the meaning of Lemma 2 and it's purpose?

---

> ### Author Response · Authors · 2025-11-22
>
> We thank the reviewer for their helpful feedback, to which we provide point-to-point answers in the following. For greater readability, we compounded comments and questions about similar topics.
>
> ---
>
> W1: Lemma 2 seems written somewhat informally and I do not follow what it is saying. In particular, it states that imposing a certain inequality "coincides with accounting for each point to hyperplane distance as ...". I don't understand precise meaning of the wording "coincides with accounting for", and hence I do not follow the meaning or significance of this lemma.
>
> W2: It's also not that clear how Lemma 2 is used in the main results. It's clear how Lemma 3 and Theorem 2 together produce the useful bound in corollary 2, but what role does Lemma 2 play in all this?
>
> Q: Can you clarify the meaning of Lemma 2 and it's purpose?
>
> ---
>
> The intent of Lemma 2 is to provide a geometric interpretation of the multi-norm constraint from the perspective of the objective function that is being minimized. Specifically, it shows that solving the variant of k-HC where the norm constraint reads
>
> $\min\{||w||_1, \sqrt{n} ||w||_\infty\} \geq 1$
>
> (we apologies for this poorly formatted formula---we could't find a way to make it render properly on OpenReview)
>
> is equivalent to solving a variant of k-HC without any norm constraints and in which the point-to-hyperplane distance (to be minimized in the objective function) is defined as
>
> $$\max\{d_\infty(a_i,H_j),\sqrt{\tfrac{1}{n}}\,d_1(a_i,H_j)\},$$
>
> that is, the maximum between the ∞-norm and a scaled 1-norm distance. This equivalence is important because it shows that the multi-norm constraint induces a mixed-distance metric that can be used to construct tighter relaxations and derive sharper theoretical bounds.
>
> In the revised version, we will restate the lemma more precisely to make this equivalence and its significance explicit.
>
> ---
>
> W2: Figures 1 and 2 could use a bit more explanation, e.g., in the caption. At first something seemed backwards, because the text explained how the feasible region of a (q,c) problem with  and  contains the feasible region for , but this appeared opposite to the pictures where a circle contained a red diamond and green square. It did not take me long to realize that this is because the feasible region is everything that lies _outside_ these colored shapes, but it would have made things a lot clearer a lot quicker to note this explicitly, rather than just stating these are illustrations of certain feasible regions. Also, in the captions, shouldn't you write 2 instead of n in ?
>
> Thanks for this. We revised the captions following your suggestions.
>
> ---
>
> W4: This is minor, but there are also some typos to fix:
>
> - lemma 1: coincide Also ---> add missing period
> - line 072: "d_p intrinsically" --> missing the word "is"?
> - line 055: "nonzero global lower bounds is" --> bound is or bounds are
>
> Thanks. We fixed these typos.

---

> > ### Comment · Reviewer_RjaT · 2025-11-25
> > **Thanks**
> >
> > Thanks for the follow up and clarifications. I maintain my positive view of the paper.

---

### Official Review · Reviewer_mEub · 2025-10-31

**Soundness:** 2
**Presentation:** 2
**Contribution:** 2
**Rating:** 4
**Confidence:** 3

**Summary:**

The authors give a spatial branch-and-bound (SBB) method to solve the k-hyperplane clustering problem (k-HC2), which minimizes the sum of squared Euclidean distances from points to their nearest hyperplane. By updating the MI-QCQP formulation with constraints from alternative p-norms (especially polyhedral norms like p = 1, $\infty$), they show that these additions improve SBB efficiency, reducing the number of nodes needed for nonzero lower bounds from exponential to linear under mild assumptions. Empirical results confirm speedups on the two listed datasets.

**Strengths:**

1. The k-Hyperplane Clustering problem (k-HC2) is a fundamental problem in classical machine learning. Addressing the k-HC2 problem using the a MI-QCQP formulation with constraints would be of interest to the community, particularly in the context of subspace clustering.

2. The main contribution of this work lies in using a multi-norm approach to estimate the lower bound for the objective value of k-HC_{2,1}, as compared to using a single polyhedral norm. In this way, the authors enable earlier pruning in the SBB procedure. Furthermore, under specific assumptions (as shown in Proposition 4), the authors provide theoretical guarantees for the use of two scaled polyhedral-norm constraints in the k-HC(2,1) formulation.

3. The proposed algorithm shows empirical speedups on datasets.

**Weaknesses:**

1. The paper does not discuss prior research in subspace clustering, which also aims to identify k subspaces that minimize the sum of squared Euclidean distances between each data point and its closest subspace [1][2][3].

2. The idea of using polyhedral norms and the SBB methods for k-HC2 problem has already been explored in several prior works (e.g., Dhyani and Liberti (2008), Amaldi & Coniglio (2013)). While this paper introduces a modification by incorporating multiple norm constraints($\ell_1$-norm and $\ell_{\infty}$-norm) for approximation purposes, the contribution may be viewed as incremental.

3. The problem remains computationally challenging for large-scale instances, and the paper does not clearly specify the time complexity of the proposed algorithm.

Reference:

[1]. Rademacher L, Vempala S, Wang G. Matrix approximation and projective clustering via iterative sampling[C]. Proceedings of the seventeenth annual ACM-SIAM symposium on Discrete algorithm, 2006, Pages 1117 - 1126.

[2]. Sohler, Christian, and David P. Woodruff. Strong coresets for k-median and subspace approximation: Goodbye dimension. 2018 IEEE 59th Annual Symposium on Foundations of Computer Science. IEEE, 2018.

[3]. Eiben, Eduard, et al. EPTAS for k-means clustering of affine subspaces. Proceedings of the 2021 ACM-SIAM Symposium on Discrete Algorithms, 2021.

**Questions:**

1. How does the proposed method compare to existing subspace clustering approaches [2-3] in terms of formulation and performance?

2. What is the time complexity of the proposed algorithm, and how does it scale with data size?

3. What are the practical applications of the k-HC² algorithm, and in which domains does it offer clear advantages?

---

> ### Author Response · Authors · 2025-11-22
>
> We thank the reviewer for their feedback, to which we provide point-to-point answers in the following. To offer clearer answers, we compounded any comments that shared a similar topic.
>
> ---
>
>
> W1: The paper does not discuss prior research in subspace clustering, which also aims to identify k subspaces that minimize the sum of squared Euclidean distances between each data point and its closest subspace.
>
> Q1: How does the proposed method compare to existing subspace clustering approaches  in terms of formulation and performance?
>
> We thank the reviewer for the comment. Indeed, the line of work on coresets for projective clustering and subspace approximation (e.g., Rademacher et al., 2006; Sohler & Woodruff, 2018; Eiben et al., 2021) provides powerful tools for reducing large datasets to small, weighted subsets while maintaining provable approximation guarantees. These results can be directly leveraged in our setting: a coreset can first be constructed to downsample the data and obtain a \((1+\varepsilon)\)-approximate instance, after which our SBB-based solver can be applied to compute a provably optimal solution on the reduced problem.
>
> This integration would yield a hybrid approach—approximate in data, exact in optimization—potentially combining the scalability of coreset methods with the global optimality guarantees of our formulation. We find this direction very promising and will explicitly mention it in the conclusion as an exciting avenue for future work, since developing such a pipeline requires further algorithmic and theoretical investigation.
>
> ---
>
> W2: The idea of using polyhedral norms and the SBB methods for k-HC2 problem has already been explored in several prior works (e.g., Dhyani and Liberti (2008), Amaldi & Coniglio (2013)). While this paper introduces a modification by incorporating multiple norm constraints (l1-norm and linf-norm) for approximation purposes, the contribution may be viewed as incremental.
>
> Our contribution is not merely the inclusion of multiple norm constraints, but rather a new theoretical and algorithmic framework that establishes, for the first time, provable complexity guarantees for these strengthened formulations.
>
> Specifically, no prior study analyzed the number of nodes required to obtain a nonzero global lower bound or provided a theoretical explanation for why certain norm constraints lead to exponentially versus polynomially sized trees. We show that the 1-norm-based strengthening yields exponential growth, whereas the infinity-norm-based version leads to polynomial behavior $O(nk)$—a sharp, formal distinction not established before.
>
> Moreover, we propose and empirically validate the multi-norm formulation, for which we also derive a theoretical approximation bound (Corollary 2). Although this formulation is tighter, it reveals new insights about solver behavior, symmetry breaking, and combinatorial complexity. These analyses clarify why certain norm-based strengthenings perform better and establish a direct connection between geometric relaxations and SBB efficiency.

---

> ### Author Response · Authors · 2025-11-22
>
> ---
>
> W3: The problem remains computationally challenging for large-scale instances, and the paper does not clearly specify the time complexity of the proposed algorithm.
>
> Q2: What is the time complexity of the proposed algorithm, and how does it scale with data size?
>
> The challenging nature of the problem for large-scale instances is inherent, as the k-HC problem is NP-hard for any choice of norm. Because our approach is an exact method, its worst-case complexity is necessarily exponential—this is unavoidable unless P=NP. That said, the focus of our analysis is on understanding and improving the structure of the search tree, which drives practical performance, when extra constraints are added to the problem's formulation. Indeed, while the Spatial Branch-and-Bound (SBB) framework we work with cannot not admit a polynomial-time guarantee for an NP-hard problem, its effectiveness depends crucially on how quickly it can obtain a nonzero global lower bound, enabling pruning. Our theoretical contribution precisely quantifies this early-phase complexity—something rarely done in the literature.
>
> Tables 3–4 illustrate how the different formulations scale across increasing instance sizes and dimensions. The trend is consistent and clear in both the LowDim and HighDim datasets: the formulation strengthened with the ∞-norm constraints consistently achieves the lowest wall-clock times, often by one or two orders of magnitude compared to the baseline one.
>
> In the LowDim dataset (Table 3), the ∞-norm formulation solves almost all instances to global optimality within seconds or minutes, while the baseline often requires hours or hits the time limit (168 000 s). Similarly, in the HighDim dataset (Table 4), the ∞-norm formulation remains tractable even as (m,n,k) grow, whereas the other variants frequently time out. Notably, it yields the highest number of solved instances and the smallest overall computation times.
>
> These results confirm that the ∞-norm strengthening not only provides the best theoretical scaling (with an $O(nk)$ node bound) but also demonstrates the best empirical scalability. It remains efficient as both the problem dimension and number of points increase, validating that our strengthened formulation substantially improves solver performance in practice.
>
> ---
>
> Q3: What are the practical applications of the k-HC2 algorithm, and in which domains does it offer clear advantages?
>
> Many practical applications of the k-HC2 algorithm are already outlined in the paper’s introduction. The problem arises when one needs to identify sets of points that are approximately co-linear or (hyper)co-planar. As stated in the paper, k-HC2 naturally arises in several domains, including:
>
> Line and surface detection in digitally sampled images and 3D environments (Amaldi & Mattavelli, 2002);
> - Medical prognosis and diagnostic modeling (Bradely & Mangasarian, 2000);
> - Linear facility location (Megiddo & Tamir, 1982);
> - Identification of piecewise affine hybrid systems (Ferrari-Trecate et al., 2003);
> - Principal or sparse component analysis (Washizawa & Cichocki, 2006; He & Cichocki, 2007; Tsakiris & Vidal, 2017);
> - Nonlinear regression (He & Qin, 2010);
> - Dictionary learning and sparse representation (Zhang et al., 2013; Georgiev et al., 2007);
> - LiDAR data classification (Kong et al., 2013).
>
> In these areas, k-HC provides clear advantages whenever the data can be well represented by multiple hyperplanes or local linear models. Compared to heuristic or spectral methods for subspace clustering, our exact Spatial Branch-and-Bound (SBB) approach guarantees global optimality and provides theoretical complexity bounds on the branching process. This makes it particularly valuable in applications where correctness and interpretability are critical, such as system identification, medical modeling, and geometric reconstruction.

---

> ### Comment · Reviewer_mEub · 2025-11-26
> **Response to the Authors**
>
> Thank you for the detailed rebuttal and additional clarifications. The response addresses some of my earlier concerns, in particular by better situating the work within the subspace clustering literature, clarifying the role and guarantees of the norm-strengthening formulation, and explaining the (necessarily exponential) complexity along with some indicative empirical scaling and application domains. However, I remain unconvinced that the methodological novelty over prior SBB/MI-QCQP approaches is substantial enough and that the practical benefits are clearly demonstrated against strong subspace-clustering baselines, especially in more challenging large-scale settings (lack of experimental supports). Therefore, I would like to maintain my original overall score.

---

### Official Review · Reviewer_oow2 · 2025-11-02

**Soundness:** 2
**Presentation:** 2
**Contribution:** 3
**Rating:** 4
**Confidence:** 4

**Summary:**

This paper introduces a method to solve the k-Hyperplane Clustering problem in the 2-norm ($k-HC_2$) to global optimality using spatial branch-and-bound (SBB). The approach strengthens the classical mixed-integer quadratically constrained quadratic programming (MI-QCQP) formulation by incorporating constraints from polyhedral norm variants $(p = 1, \infty)$. The authors show that including ∞-norm constraints allows the SBB method to obtain a nonzero lower bound in O(nk) nodes, compared to $\Omega(2^{k(n−1)})$ nodes for the baseline. Experiments on synthetic datasets report speedups of 8–41 times, improving the number of instances solved to global optimality.

**Strengths:**

1. Theoretical analysis proves that adding polyhedral norm constraints reduces the number of SBB nodes required to obtain a nonzero lower bound from exponential to polynomial.
2. The method is evaluated on two synthetic testbeds (Low-dim and High-dim) with statistical tests (Wilcoxon signed-rank) confirming significant speedups.
3. Clear MI-QCQP and MILP formulations are provided for the polyhedral norm constraints.
4. The multi-norm relaxation framework is general and may be applied to other problems with nonconvex norm constraints.

**Weaknesses:**

1. Experiments are limited to small-scale instances (up to m = 30, n = 5, k = 5) and use synthetically generated data with a specific noise model.
2. The theoretical advantage of $\infty$-norm constraints assumes branching occurs first on the binary variables of the polyhedral norm formulation; performance under default branching strategies is not tested.
3. The analysis focuses only on the number of nodes to the first nonzero bound, not the total tree size or gap closure.
4. No comparison with specialized heuristics or state-of-the-art approximate methods is provided.
5. Validation is limited to synthetic data; real-world applicability is not assessed.
6. The method requires solving complex MI-QCQPs, which may not scale to very large instances.
7. There is also no analysis of the trade-off between solution quality and computation time for practical use.
8. The paper does not provide guidance on selecting which polyhedral norm constraints to include for best performance.

**Questions:**

1. How does the method scale with the number of data points m, given that the assignment problem may become the bottleneck?
2. Can the authors provide wall-clock time profiles and total node counts (not just medians) to better characterize tree size and convergence?
3. How robust are the speedups under solver-default branching rules and parameters?
4. Why does the multi-norm formulation perform worse than the individual $\infty$-norm formulation in some cases?

---

> ### Author Response · Authors · 2025-11-22
>
> We thank the reviewer for their feedback, to which we provide point-to-point answers in the following. To offer clearer answers, we compounded any comments that shared a similar topic.
>
> ---
>
> W1: Experiments are limited to small-scale instances (up to m = 30, n = 5, k = 5) and use synthetically generated data with a specific noise model.
>
> W5: Validation is limited to synthetic data; real-world applicability is not assessed.
>
> The benchmark we used is the standard one that has been used in the literature (Amaldi, Coniglio 2013; Dhyani, Liberti 2008) for evaluating solution methods for k-hyperplane clustering. These benchmarks are standard and allow controlled instance generation and reproducibility, and allow verifying global optimality. As shown in (Amaldi, Coniglio 2013; Dhyani, Liberti 2008) and confirmed in our own experiments, this benchmark contains very hard to solve instances. The sizes we consider are directly not only aligned, but also larger, than those used in previous exact works on the problem. Exact global optimization (the setting in which we work in this paper) requires instances where the true optimum is verifiable, which is generally impossible on larger instances where no method can certify optimality within reasonable time. Also note that, with our algorithm, we managed to show that all the heuristic solutions found in Amaldi & Coniglio (2013) on the 24 instances therein considered were optimal.
>
> ---
>
> W2: The theoretical advantage of ∞-norm constraints assumes branching occurs first on the binary variables of the polyhedral norm formulation; performance under default branching strategies is not tested.
>
> The assumption is motivated by the fact that modern SBB solvers tend to almost always branch on 0-1 variables. Notice, though, that our tests were run entirely under default branching strategies and default parameters. We didn't force Gurobi in any way to branch on the 0-1 vars first.
>
> ---
>
> W3: The analysis focuses only on the number of nodes to the first nonzero bound, not the total tree size or gap closure.
>
> We focused our theoretical analysis on the number of nodes required to obtain the first nonzero global lower bound, as this milestone fundamentally determines when the SBB algorithm begins to make progress. Before this point, the bound remains identically zero—a degenerate bound given that the objective is a sum of squares—meaning that no pruning can occur anywhere in the tree. Hence, the number of nodes to the first nonzero bound is directly indicative of how quickly the solver transitions from a trivial to an informative search.
>
> It is worth emphasizing that rigorous results of this type are exceptionally rare in the branch-and-bound and spatial branch-and-bound literature. Deriving explicit complexity guarantees for nonlinear mixed-integer formulations is notoriously difficult; our analysis provides one of the few instances where such a closed-form bound on node count can be established. The results—particularly the $O(nk)$ bound for the ∞-norm formulation—offer an interpretable and theoretically grounded measure of algorithmic efficiency.
>
> Finally, as stated in the paper, the k-HC problem is NP-hard regardless of the chosen norm. This directly implies that no polynomial bound can exist on the total number of nodes or on complete gap closure unless  P = NP. Our analysis therefore targets the most meaningful and provable performance criterion: the point at which the search tree first becomes nontrivial and pruning becomes theoretically possible.
>
> ---
>
> W4: No comparison with specialized heuristics or state-of-the-art approximate methods is provided.
>
> We do not compare against heuristic or approximate methods because our goal is to study global optimality and the behavior of exact SBB formulations. Heuristics are not designed to find globally optimal solutions and, therefore, are not comparable in terms of certified optimality or lower-bound quality. While such methods can provide good incumbents, such incumbents would not affect the strength of the relaxation or the speed at which a nonzero global bound is obtained.
>
> ---
>
> ---
>
> W6: The method requires solving complex MI-QCQPs, which may not scale to very large instances.
>
> Exact k-HC₂ is intrinsically hard, and any globally optimal approach must ultimately solve a MI-QCQP. Our contribution does not change this fundamental complexity, but it substantially improves the efficiency of the exact solver by strengthening the relaxation so that a useful global bound appears orders of magnitude earlier. This reduces the effective search dramatically, as confirmed in our experiments.

---

> ### Author Response · Authors · 2025-11-22
>
> W7: There is also no analysis of the trade-off between solution quality and computation time for practical use.
>
> For our method, there is no trade-off between solution quality and computation time only because the algorithm always returns the globally optimal solution and the strengthened formulations do not introduce any approximation. The primary purpose of our methods is to improve the certification (lower-bound) side of the search. This is, arguably, the crucial goal of any (spatial) branch and bound method: providing a computational proof of optimality of a solution that is ether given or found within the tree. Previous works show that finding optimal solutions heuristically is a rather easy task. Proving their optimality, on the contrary, is a very hard task, and improving such a hard task is what we set out to do in this paper.
>
> ---
>
> W8: The paper does not provide guidance on selecting which polyhedral norm constraints to include for best performance.
>
> We would like to note that this point is already discussed in the paper. In Section 4.4 (“Strengthened Formulations”) and in the discussion following Propositions 3–4, we explain which polyhedral norm constraints are preferable in practice. The paper shows that, when solving the 2-norm problem, including the ∞-norm constraint generally offers the best computational performance—achieving linear branching complexity \(O(nk)\) and faster runtimes—whereas the 1-norm and multi-norm constraints, while theoretically tighter, introduce additional binary variables and branching overhead. As such, we recommend using the ∞-norm constraint when the 2-norm constraint is already imposed, and reserving the multi-norm formulation for cases where the 2-norm constraint is not included. We will make this guidance clearer in the final version.
>
> ---
>
> Q1: How does the method scale with the number of data points m, given that the assignment problem may become the bottleneck?
>
> Scaling with respect to the number of data points \(m\) can be assessed from Tables 2–3. We reproduce below a representative subset obtained with \(n=2\) and \(k=3\), showing computing time as a function of \(m\) for the four formulations considered. While the assignment subproblem will eventually become the bottleneck as \(m\) grows, our results indicate that the norm constraints dominate the computational burden much earlier. In particular, any formulation including a polyhedral norm achieves dramatically smaller solving times than the one relying solely on the 2-norm constraints, showing that the strengthened formulations reach non-zero bounds and prune the tree far more efficiently.
>
> | **m** | **(k–HC(2,1))** | **(k–HC(2,1),(∞,1))** | **(k–HC(2,1),(1,1/√n))** | **(k–HC(multi,1))** |
> |:----:|:---------------:|:---------------------:|:-------------------------:|:------------------:|
> | 10 | 0.7 | 1.0 | 0.8 | 1.0 |
> | 14 | 31.9 | 4.4 | 3.4 | 5.4 |
> | 18 | 488.9 | 3.9 | 4.4 | 4.6 |
> | 22 | 2213.3 | 11.2 | 11.2 | 9.8 |
> | 25 | 168 000.0 | 936.6 | 96.1 | 221.0 |
> | 26 | 168 000.0 | 39.2 | 56.6 | 28.3 |
> | 27 | 168 000.0 | 1678.4 | 2687.7 | 238.6 |
> | 28 | 168 000.0 | 293.1 | 471.3 | 153.5 |
> | 29 | 168 000.0 | 7694.9 | 6029.0 | 1476.4 |
> | 30 | 168 000.0 | 172.9 | 191.2 | 44.3 |

---

> ### Author Response · Authors · 2025-11-22
>
> Q2: Can the authors provide wall-clock time profiles and total node counts (not just medians) to better characterize tree size and convergence?
>
> Let us clarify that wall-clock times are already reported in Tables 2–3 of the paper. To complement this, we have now included the total node counts in a new Table 3 for the HighDim instances, which represent the most challenging settings. This addition provides a clearer picture of the relative tree sizes and convergence behavior across formulations.
>
> | **m** | **n** | **k** | **n2 (k–HC(2,1))** | **n2nI (k–HC(2,1),(∞,1))** | **n2n1 (k–HC(2,1),(1, 1/√n))** | **n2n1nI (k–HC(multi,1))** |
> |:----:|:----:|:----:|:-------------:|:-------------:|:-------------:|:-------------:|
> | 10 | 2 | 4 | 38 392 | 16 357 | 10 301 | 23 588 |
> | 10 | 4 | 2 | 17 033 | 4 739 | 18 112 | 16 937 |
> | 11 | 2 | 4 | 78 168 | 29 502 | 17 868 | 41 565 |
> | 11 | 2 | 5 | 4 654 600 | 1 151 890 | 958 992 | 8 332 930 |
> | 11 | 4 | 2 | 21 404 | 8 701 | 6 989 | 8 584 |
> | 12 | 2 | 4 | 287 736 | 45 100 | 22 786 | 75 515 |
> | 12 | 2 | 5 | 1 228 060 | 440 626 | 169 063 | 687 073 |
> | 12 | 4 | 2 | 61 743 | 5 820 | 19 455 | 26 611 |
> | 12 | 5 | 2 | 85 702 | 42 506 | 36 266 | 60 555 |
> | 13 | 2 | 4 | 791 227 | 52 720 | 41 656 | 99 091 |
> | 13 | 2 | 5 | 2 621 440 | 347 674 | 154 070 | 383 310 |
> | 13 | 3 | 4 | 11 864 400 | 19 678 500 | 23 810 300 | 18 600 100 |
> | 13 | 4 | 2 | 41 063 | 19 382 | 9 097 | 24 744 |
> | 13 | 4 | 3 | 2 629 260 | 1 284 080 | 1 646 530 | 9 724 370 |
> | 13 | 5 | 2 | 139 309 | 24 034 | 40 346 | 48 397 |
> | 14 | 2 | 4 | 2 355 780 | 61 667 | 34 681 | 166 633 |
> | 14 | 2 | 5 | 19 826 000 | 2 014 470 | 715 573 | 1 582 970 |
> | 14 | 3 | 4 | 21 011 800 | 5 928 130 | 6 446 500 | 16 330 600 |
> | 14 | 4 | 2 | 197 555 | 10 637 | 15 814 | 19 679 |
> | 14 | 4 | 3 | 3 653 010 | 1 509 690 | 1 914 490 | 14 396 900 |
> | 14 | 5 | 2 | 367 215 | 34 060 | 40 631 | 67 349 |
> | 15 | 2 | 4 | 4 819 300 | 88 293 | 63 762 | 142 499 |
> | 15 | 2 | 5 | 15 129 700 | 815 961 | 240 698 | 1 057 150 |
> | 15 | 3 | 4 | 20 797 200 | 6 821 170 | 4 221 290 | 17 132 800 |
> | 15 | 4 | 2 | 123 055 | 14 603 | 21 152 | 19 678 |
> | 15 | 4 | 3 | 9 399 350 | 1 068 560 | 949 518 | 4 432 280 |
> | 15 | 5 | 2 | 285 279 | 25 590 | 83 182 | 63 958 |
> | 16 | 2 | 4 | 20 072 700 | 387 933 | 86 122 | 177 715 |
> | 16 | 2 | 5 | 18 348 500 | 1 839 480 | 977 328 | 1 834 550 |
> | 16 | 3 | 4 | 18 615 700 | 12 285 100 | 5 353 010 | 16 241 100 |
> | 16 | 3 | 5 | 16 743 900 | 16 840 700 | 15 925 600 | 14 859 600 |
> | 16 | 4 | 2 | 622 968 | 29 784 | 34 762 | 25 049 |
> | 16 | 4 | 3 | 12 307 300 | 1 297 660 | 1 169 750 | 4 721 850 |
> | 16 | 5 | 2 | 1 411 490 | 66 069 | 100 491 | 107 740 |
> | 17 | 2 | 4 | 23 783 300 | 108 023 | 120 569 | 176 621 |
> | 17 | 2 | 5 | 18 531 600 | 4 274 570 | 3 036 550 | 3 629 780 |
> | 17 | 3 | 4 | 18 137 400 | 10 911 800 | 5 561 730 | 16 517 600 |
> | 17 | 3 | 5 | 16 997 500 | 15 721 000 | 16 018 000 | 13 877 100 |
> | 17 | 4 | 2 | 599 093 | 22 341 | 26 660 | 18 985 |
> | 17 | 4 | 3 | 18 403 800 | 2 103 800 | 2 034 520 | 7 360 030 |
> | 17 | 4 | 4 | 16 580 800 | 14 203 900 | 13 111 500 | 12 754 700 |
> | 17 | 5 | 2 | 3 190 660 | 187 282 | 217 843 | 189 688 |
> | 17 | 5 | 3 | 16 600 600 | 15 857 000 | 12 400 800 | 13 129 400 |
>
> The pattern clearly confirms the theoretical analysis. (∞,1) consistently yields the smallest trees, matching the \(O(nk)\) node complexity predicted by Proposition 4. The (1,1/√n) and multi-norm variants are tighter relaxations but can expand larger trees due to extra binaries and interplay between 0-1 variables which may delay branching in full on one of the norms. The baseline 2-norm model scales exponentially, reaching millions of nodes even for modest dimensions.

---

> ### Author Response · Authors · 2025-11-22
>
> Q3: How robust are the speedups under solver-default branching rules and parameters?
>
> Our experiments were entirely run under solver-default parameters. We run some preliminary experiments with different values of the `mipemphasis` parameter (focusing more on primal or dual bounds), but found no measurable difference in performance with those.
>
> ---
>
> Q4: Why does the multi-norm formulation perform worse than the individual-norm formulation in some cases?
>
> While the multi-norm formulation is tighter (better approximation factor), it can be slower because it inherits the exponential branching burden of the 1-norm part and adds more binaries while relaxing symmetry breaking. The ∞-norm-only strengthening avoids that pitfall, reaches a non-zero lower bound in $O(nk)$ nodes, and therefore tends to be faster in practice even if its relaxation is weaker than the multi-norm one.  These mechanisms and the observed runtimes align with Propositions 3–4 and Tables 1–2 in the paper. Keep in mind that we are adding these polyhedral norm constraints on top of the 2-norm ones. Our results suggest that employing the ∞-norm-only constraints is extremely helpful to drive the bound off of zero, after which one could continue branching in a classical spatial branch-and-bound fashion on the continuous variables involved in nonlinear constraints (i.e., the $w$ variables). In a way, this shows that it is ideal to introduce a single device capable of moving the bound away from zero as soon as possible which requires as few branching operations as possible, which is precisely what the ∞-norm constraints do. In particular, since we do not impose an priority on which 0-1 variables branching should take place on, it's likely that Gurobi will branch on a mixture of 1-norm and ∞-norm constraints variables before having completed branching on either of them. This can result obviously delay the point in which a nonzero lower bound is obtained, ultimately reducing the solver's performance.

---

### Author Response · Authors · 2025-12-02

Dear Area Chair,

Thank you for handling our paper. We are thankful to the reviewers for looking into our work.

Across the reviews, there is a clear shared recognition of the paper’s technical depth and relevance.

Reviewer RjaT emphasizes that “the mathematical contributions of the paper is non-trivial; the technical contribution of the paper is quite high” and that “even though the paper is proving a large number of non-trivial technical results, the presentation is very good and explains the main ideas and components (and how they fit together) very well.” They further note that our “approach leads to concrete improvements in numerical experiments.”

Reviewer oow2 highlights our main theoretical contribution: “Theoretical analysis proves that adding polyhedral norm constraints reduces the number of SBB nodes required to obtain a nonzero lower bound from exponential to polynomial.” They also stress that “the multi-norm relaxation framework is general and may be applied to other problems with nonconvex norm constraints”, and that our approach leads to “speedups of 8–41 times, improving the number of instances solved to global optimality.”

Reviewer mEub underlines the problem’s importance and the core idea behind our approach: “The k-Hyperplane Clustering problem (k-HC2) is a fundamental problem in classical machine learning,” and “the main contribution of this work lies in using a multi-norm approach to estimate the lower bound for the objective value of k-HC_{2,1}, [so that we] enable earlier pruning in the SBB procedure.” They also agree that “the proposed algorithm shows empirical speedups on datasets.”

The main reservations raised by Reviewers oow2 and mEub concern (i) the experimental setting (synthetic, small to medium-scale instances, no direct comparison to subspace-clustering heuristics), (ii) questions on complexity and scalability, and (iii) clarity on some technical points and positioning relative to the subspace-clustering literature. During the rebuttal, we responded point-by-point and made several concrete additions and clarifications:

* On experimental scale and benchmark choice (oow2 W1/W5), we clarified that we use the standard exact k-hyperplane clustering benchmarks from the literature (Amaldi & Coniglio, 2013; Dhyani & Liberti, 2008), which are known to be very challenging, and that the sizes we consider are directly not only aligned with, but also larger than those used in previous exact works on the problem. We also pointed out that, with our method, we managed to show that all the heuristic solutions found in Amaldi & Coniglio (2013) on the 24 instances therein considered were optimal. This shows that the experimental setup we considered is grounded in the k-HC literature. It also underscored that one of the contributions of our work is that our method leads to globally optimal solutions and can be used to certify global optimality on hard instances where good solutions were available but a proof of their optimality was lacking.

* On branching assumptions and solver robustness (oow2 W2/Q3), we clarified that our theoretical analysis assumes branching on the 0–1 variables of the polyhedral norm formulation, which is aligned with typical SBB practice. Importantly, in the experiments our tests were run entirely under default branching strategies and default parameters. We didn't force Gurobi in any way to branch on the 0–1 vars first, and we reported that preliminary experiments with different values of the mipemphasis parameter found no measurable difference in performance. This supports the practical robustness of our observed speedups. Also, please note that our method is entirely devoid of parameter tuning. We used default parameters in all of our experiments.

* On tree size and convergence (oow2 W3/Q2), we added new experimental results. We emphasized that wall-clock times were already in the main tables, and included the total node counts in a new Table 3 for the HighDim instances, giving a detailed picture of tree sizes. The new table shows that the infinity-norm-strengthened formulation dramatically reduces node counts, consistently matching the predicted (O(nk)) behavior: The pattern clearly confirms the theoretical analysis, with the formulation featuring infinity-norm constraints yielding the smallest trees, matching the (O(nk)) node complexity predicted by Proposition 4. The baseline 2-norm model scales exponentially, reaching millions of nodes even for modest dimensions.

---

> ### Author Response · Authors · 2025-12-02
>
> * On complexity and scalability (mEub W3/Q2, oow2 W6), we stressed that the k-HC problem is NP-hard for any norm and that any exact method must have exponential worst-case complexity unless P=NP. Our contribution is therefore to precisely quantify early-phase complexity of computing a nonzero lower bound, which we achieve by proving bounds on the number of nodes until the first nonzero global lower bound is computed, a milestone after which pruning becomes possible. The experiments confirm that the infinity-norm strengthening both enjoys the best theoretical scaling and remains efficient as both the problem dimension and number of points increase. Let us reiterate that results of this type are extremely rare in integer programming (let alone nonlinear integer programming).
>
> * On the relation to subspace clustering and coresets (mEub W1/Q1), we explicitly connected our work to the projective clustering and coreset literature. Indeed, the line of work on coresets for projective clustering and subspace approximation provides a tool for reducing large datasets to small, weighted subsets while maintaining provable approximation guarantees. We pointed out that such coresets can be directly leveraged in our setting, as a coreset can first be constructed to downsample the data, after which our SBB-based solver can be applied to compute a provably optimal solution on the reduced problem. We proposed this as a promising hybrid approximate-in-data, exact-in-optimization pipeline, and mentioned it in the revised version of our paper.
>
> * On novelty over prior norm/SBB formulations (mEub W2), we clarified that our contribution is not just “adding multiple norms” but a new theoretical and algorithmic framework which, for the first time, provides complexity guarantees for these strengthened formulations. In particular, no prior study analyzed the number of nodes required to obtain a nonzero global lower bound or provided a theoretical explanation for why certain norm constraints lead to exponentially versus polynomially sized trees. We proved that 1-norm strengthening yields exponential growth, whereas infinity-norm strengthening yields polynomial behavior, a sharp, formal distinction not established before. We also provided approximation guarantees for the multi-norm formulation (Corollary 2) and analyzed how tighter relaxations interact with branching and symmetry, giving new insight into solver behavior. We reiterate that such results are not only entirely new, but very rare to come by in the integer programming literature.
>
> * On clarity of Lemma 2 and figures (RjaT W1–W2), we clarified that the intent of Lemma 2 is to give a geometric interpretation of the multi-norm constraint: solving the multi-norm variant of k-HC is equivalent to solving another variant of k-HC without any norm constraints and in which the point-to-hyperplane distance is defined as the maximum between the infinity-norm and a scaled 1-norm distance. We restated the lemma more precisely and revised the captions of Figures 1 and 2 as requested.
>
> * On practical applications and domains (mEub Q3), we reiterated the list of areas where k-HC2 naturally appears, which were already mentioned in detail in our paper.
>
> In light of the reviewers’ recognition that the paper offers non-trivial mathematical contributions, a clear and well-presented theoretical framework, and concrete empirical gains in exact global optimization of k-HC2, together with the additional results and clarifications provided in the rebuttal (especially the new node-count statistics and expanded discussion of complexity and connections to subspace clustering), we believe the revised version would be of genuine interest to the ICLR community at the intersection of optimization and machine learning.
>
> Best regards,
> The Authors

---

### Meta-Review · Area_Chair_J3TY · 2026-01-05

**Summary:**

1. The algorithm is developed based on SBB and MI-QCQP. Compared with prior work, the novelty appears insufficient.
2. The runtime for a large dataset is significant.
3. Technical concerns about the proposed algorithm, including the analysis of the number of nodes to the first nonzero bound

**Reviewer Concerns:**

The rebuttal addresses technical concerns and clarifies that the focus is on exact algorithms, which are not expected to work for large datasets.

Reviewer mEUb still has concerns about the novelty. I agree that the study on the impact of different types of constraints may sound less significant. But I may appreciate more on the clarify of a fundamental idea with outperformed empirical performance.

**Reviewer Scores:**

No change

---

### Decision · Program_Chairs · 2026-01-26

Accept (Poster)